# Consensus molecular subtype differences linking colon adenocarcinoma and obesity revealed by a cohort transcriptomic analysis

**Michael W. Greene**[1]*, **Peter T. Abraham**[2], **Peyton C. Kuhlers**[1,3], **Elizabeth A. Lipke**[2], **Martin J. Heslin**[4], **Stanley T. Wijaya**[1], **Ifeoluwa Odeniyi**[1]

**1** Department of Nutrition, Dietetics and Hospitality Management, Auburn University, Auburn, AL, United States of America, **2** Department of Chemical Engineering, Auburn University, Auburn, AL, United States of America, **3** Department of Biostatistics, University of North Carolina at Chapel Hill, Chapel Hill, NC, United States of America, **4** Mitchell Cancer Institute, The University of South Alabama, Mobile, AL, United States of America

* mwgreene@auburn.edu

**Data Availability Statement:** All relevant data are within the paper and its Supporting Information files.

## Abstract

Colorectal cancer (CRC) is the third-leading cause of cancer-related deaths in the United States and worldwide. Obesity—a worldwide public health concern—is a known risk factor for cancer including CRC. However, the mechanisms underlying the link between CRC and obesity have yet to be fully elucidated in part because of the molecular heterogeneity of CRC. We hypothesized that obesity modulates CRC in a consensus molecular subtype (CMS)-dependent manner. RNA-seq data and associated tumor and patient characteristics including body weight and height data for 232 patients were obtained from The Cancer Genomic Atlas–Colon Adenocarcinoma (TCGA-COAD) database. Tumor samples were classified into the four CMSs with the CMScaller R package; body mass index (BMI) was calculated and categorized as normal, overweight, and obese. We observed a significant difference in CMS categorization between BMI categories. Differentially expressed genes (DEGs) between obese and overweight samples and normal samples differed across the CMSs, and associated prognostic analyses indicated that the DEGs had differing associations on survival. Using Gene Set Enrichment Analysis, we found differences in Hallmark gene set enrichment between obese and overweight samples and normal samples across the CMSs. We constructed Protein-Protein Interaction networks and observed differences in obesity-regulated hub genes for each CMS. Finally, we analyzed and found differences in predicted drug sensitivity between obese and overweight samples and normal samples across the CMSs. Our findings support that obesity impacts the CRC tumor transcriptome in a CMS-specific manner. The possible associations reported here are preliminary and will require validation using in vitro and animal models to examine the CMS-dependence of the genes and pathways. Once validated the obesity-linked genes and pathways may represent new therapeutic targets to treat colon cancer in a CMS-dependent manner.

**Funding:** The National Center for Advancing Translational Research of the National Institutes of Health (NIH) https://ncats.nih.gov/ (UL1TR003096-01, MG and EL) and the United States Department of Agriculture, National Institute of Food and Agriculture (NIFA) Hatch Grant https://nifa.usda.gov/program/hatch-act-1887-multistate-research-fund (ALA044-1-18037, MG). The funders had no role in study design, data collection and analysis, decision to publish, or preparation of the manuscript.

**Competing interests:** The authors have declared that no competing interests exist.

## Introduction

Improvements in colorectal cancer (CRC) screening, diagnosis, advanced surgical techniques, and preoperative and postoperative treatment have led to reduced CRC incidence and mortality [1]. Yet, CRC incidence rates remain high in states with a high prevalence of obesity [2]. CRC remains the third most common non skin-related cancer and the third leading cause of cancer-related mortality in the United States [3]. In addition to obesity, smoking, an unhealthy diet, high alcohol consumption, and physical inactivity are well-known risk factors that are potentially preventable [4].

Based on bulk transcriptomics, 4 consensus and 1 unclassified CRC consensus molecular subtypes have been proposed (CMS1-4) [5, 6]. CMS1 has been termed a microsatellite instability (MSI) immune subtype based on the clustering of MSI tumors and the strong infiltration of immune cells. Worse survival after relapse is associated with patients with CMS1 tumors [6]. CMS2 and CMS4 –the two most prevalent molecular subtypes of CRC–represent 37% and 23% of early-stage CRC tumors, respectively [6, 7]. The CMS2 subtype has been termed 'canonical' due to the upregulation of classical CRC pathways first proposed by Fearon and Vogelstein [8]. The CMS3 subtype has been termed the metabolic subtype due to the metabolic dysregulation at the transcriptome level. Lastly, CMS4 has been termed 'mesenchymal' due to activation of epithelial–mesenchymal transition (EMT) and overexpression of proteins implicated in ECM remodeling and complement signaling thought to be mediated by a stromal-enriched inflamed microenvironment [5]. There is a significantly higher risk of distant relapse and death for patients diagnosed with early-stage CMS4 [6].

The epidemiological evidence linking CRC with obesity, and its associated pathophysiological metabolic state is strong [9–12]. Even though colon and rectal cancer are grouped together as CRC, abdominal obesity and the metabolic syndrome are more strongly linked to colon cancer than rectal cancer [9, 11]. Obesity and the associated pathophysiological conditions of insulin resistance and inflammation afflict approximately a third of the adult population in the United States [13, 14]. The pathophysiological state associated with obesity includes visceral adipose tissue and hepatic dysfunction which leads to systemic insulin resistance and inflammation, the dysregulation of adipokines, and dysbiosis (microbial imbalance) [15, 16]; all of these pathophysiological alterations have been hypothesized to promote a favorable niche for the pathogenesis of CRC [17–21]. Insulin resistant visceral adipose tissue participates in crosstalk with CRC and promotes a favorable niche by secreting metabolites, growth factors, and proinflammatory cytokines [22–26]. Elevated pro-inflammatory cytokines are associated with an increased risk of CRC [9, 27–30], and a circulating inflammatory signature (high miR-21, IL-6, and IL-8) predicts lower progression-free and overall survival of patients with metastatic CRC [31]. The obese pathophysiological state of insulin resistance and inflammation have been shown to stimulate CRC tumor growth in animal models [32–39].

Thus, there is compelling epidemiological and experimental evidence linking obesity to CRC [9–12]–although more strongly for colon cancer [9, 11]. The obesity-cancer link is thought to be driven by multiple obesity-derived factors that activate pathways mediating cell signaling, proliferation, and tumor progression [40]. Yet, there does not exist a framework for activation of these obesity-driven cell pathways. Thus, we questioned whether the effect of obesity on cell signaling, proliferation, and tumor progression pathways in CRC tumors is dependent on the CMS of the tumor. Therefore, we undertook a study to examine the transcriptomic profile of colon adenocarcinoma tumors from obese patients compared to healthy and overweight BMI patients to determine whether obesity modulates cell signaling, proliferation, and tumor progression pathways in a similar manner across the four CMSs. We also examined whether prognostic survival outcomes and predicted drug response in obesity

associated differentially expressed genes (DEGs) is similar between the four CMSs. Our secondary objective was to determine whether the transcriptomic profile of tumors from overweight patients compared to healthy BMI patients is similar between the four CMSs. The knowledge gained from our findings can be used to test in vitro and in animal models whether key genes and pathways link obesity in a CMS-dependent manner to identify new therapeutic targets to treat colon cancer,

## Materials and methods

### Patients

Our approach is shown in Fig 1. RNA-seq data (HT-Seq counts) and associated tumor and patient characteristics for 454 patients were obtained from The Cancer Genomic Atlas–Colon Adenocarcinoma (TCGA-COAD) database using the R package TCGAbiolinks [41] which extracts and collates data from the Genomic Data Commons [42]. The patient characteristics of sex, age, ethnicity, and race were obtained from all 454 patients. Body weight and height were available for only 232 patients. Body mass index (BMI) was calculated, and each patient was categorized as underweight, (BMI < 18.5) normal (BMI 18.5–24.9), overweight (BMI 25.0–29.9), and obese (BMI >30.0). The following tumor characteristics were obtained for all patients: tumor location, tumor stage (classified as Stage I–IV), number of lymph nodes examined, and number of positive lymph nodes. Lymph node ratio (LNR) was calculated as the relation of tumor-infiltrated to total examined lymph nodes and classified as LNR0 –LNR4 based on the cut-off values 0.17, 0.41, and 0.69 [43]. Patient tumor samples from the TCGA-COAD were assigned to a CMS [6] using the R package CMScaller [44], which utilizes a nearest-template prediction algorithm [45]. Twenty-three patient samples (10%) with an FDR greater than 0.05 were not assigned to a CMS. The study was approved by the Auburn University Institutional Review Board (#20–509 EX 2010).

### Transcriptomic analysis

The R package DESeq2 [46] was used to assess differential gene expression. Raw counts and associated phenotypes were inputted into DESeq2, and the following contrasts were made within each CMS category: obese vs. normal, obese vs. overweight, and overweight vs. normal. Additionally, the effect of obesity between the subtypes was evaluated by adding an interaction term to the design, which allowed for comparison between individual CMSs using specified contrasts and for comparison across the CMSs using a likelihood ratio test. Genes with a base mean expression greater than ten and an adjusted p-value less than 0.05 were used for downstream analysis and visualization Volcano plots were generated using the R package EnhancedVolcano [https://github.com/kevinblighe/EnhancedVolcano]. Gene overlap between comparisons were visualized using Euler diagrams and upset plots from the R packages eulerr [https://github.com/jolars/eulerr] and UpsetR, respectively.

Gene Set Enrichment Analysis (GSEA) was performed using the desktop GSEA software (version 4.0.3) from the Broad Institute [47, 48]. Normalized RNA-seq counts obtained from DESeq2 were used. Permutation type was set to 'gene_set,' and Human Ensembl was selected as the CHIP platform. Hallmark gene sets [49] of well-defined biological states and processes (version 7.2) were assessed in the obese vs. normal, obese vs. overweight, and overweight vs. normal comparisons for each CMS category. Only gene sets with a false discovery rate q-value less than 0.05 were reported. The normalized enrichment score (NES) was reported for the gene set.

To construct a protein-protein interaction (PPI) network from DESeq2-obtained DEGs, the Search Tool for the Retrieval of Interacting Genes (STRING; version 10.0; string-db.org)

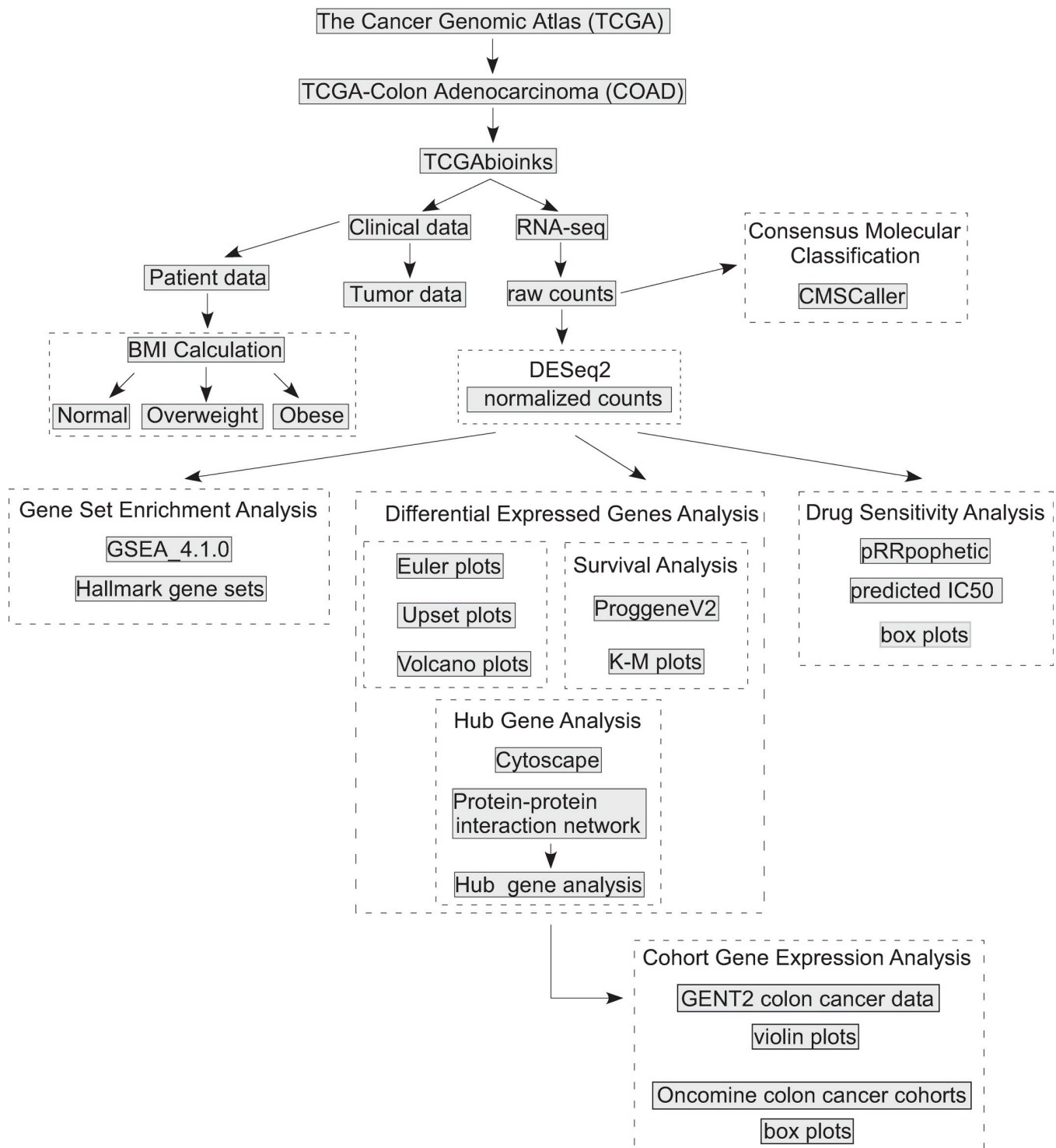

**Fig 1. Transcriptomic analysis flow chart.** Colon adenocarcinoma RNA-seq data (HT-Seq counts) and associated tumor and patient characteristics were obtained from The Cancer Genomic Atlas–Colon Adenocarcinoma (TCGA-COAD) database using the R package TCGAbiolinks. Body mass index (BMI) was calculated, and each patient was categorized as normal, overweight, and obese. Patient tumor samples from the TCGA-COAD were assigned to a consensus molecular subtype (CMS) using the R package CMScaller. Raw counts and associated phenotypes were inputted into DESeq2, and the following comparisons were made: obese vs. normal, obese vs. overweight, and overweight vs. normal for each CMS category. Volcano plots were generated using the R package EnhancedVolcano. Gene overlap between comparisons were visualized using Euler diagrams and upset plots from the R packages eulerr and UpsetR, respectively. To examine prognostic patient outcomes (Survival Analysis) DESeq2-obtained DEGs in the BMI comparisons for each CMS category was assessed using the PROGgeneV2 tool and Kaplan-Meier plots were generated to assess overall survival and relapse-free survival. Normalized RNA-seq counts obtained

from DESeq2 were used for Gene Set Enrichment Analysis (GSEA). DESeq2-obtained DEGs were used to construct a protein-protein interaction (PPI) network from the Search Tool for the Retrieval of Interacting Genes (STRING) using Cytoscape. The Cytohubba package in Cytoscape was used to perform the hub gene analysis. The hub genes were queried for mRNA expression using the aggregate microarray data using GENT2 and in colon cancer cohorts in the Oncomine database. The pRRophetic R package was used with normalized RNA-seq counts from normal, overweight, and obese patient tumors to estimate the half maximal inhibitory concentration (IC50) of drugs from the Cancer Genomic Project.

online database for PPI network construction [50] was used in Cytoscape (v3.7.1, National Resource for Network Biology, https://cytoscape.org/) [51], a bioinformatics platform for visualizing molecular interaction networks. The Cytohubba package in Cytoscape was used to perform the hub gene analysis [52] for the obese vs. normal, obese vs. overweight, and overweight vs. normal comparisons for each CMS category. Hub genes were identified using the maximal clique centrality (MCC) topological algorithm to obtain the top 10 ranked genes in all modules. A sensitivity analysis was performed using four other topological algorithms (Degree, Edge Percolated Component (EPC), Maximum Neighborhood Component (MNC), Density of Maximum Neighborhood Component (DMNC)) to identify common hub genes. Hub genes were queried for mRNA expression using the Gene Expression database of Normal and Tumor tissues 2 (GENT2) (http://gent2.appex.kr.) [53] and the Oncomine database (https://www.oncomine.org/) [54]. Colon normal tissue (n = 397) and cancer (n = 3775) microarray data collected from the NCBI GEO public database generated using the Affymetrix U133Plus2 platform was used to assess gene expression using GENT2. Violin plots were used to display the distribution of the data. The violin plots were generated using the ggplot2 library in R Studio. We examined gene expression in colon adenocarcinoma versus normal patient samples from the: 1) Hong Colorectal cohort (normal, n = 12; colon adenocarcinoma, n = 70) from Singapore [55]; 2) Skrzypczak Colorectal cohort (normal, n = 24; colon adenocarcinoma, n = 36) from Poland [56], and Kaiser Colorectal cohort (normal, n = 5; colon adenocarcinoma, n = 41) from the United States [57] using the Oncomine database. Box plots were used to display the distribution of the data.

To examine prognostic patient outcomes the expression of the top 20 significantly upregulated DESeq2-obtained DEGs in the obese vs. normal, obese vs. overweight, and overweight vs. normal comparisons for each CMS category was assessed using the PROGgeneV2 tool was used [58, 59]. We choose to use 20 genes because an approximately 20 gene set can distinguish CRC patients with low or high risk of disease relapse [60] and a 20 gene set has prognostic value for overall survival in CRC patients when adjusted for age, gender, and stage [61]. We selected the CRC cohorts with patients from the United States for which age, gender, and tumor stage were available: GSE17536 [62] and GSE41258 [63] for overall survival and GSE14333 [64] and GSE17536 [62] for relapse-free survival. All cohorts were adjusted for age, stage, and gender covariates and bifurcated based on median expression. The hazard ratios, 95% confidence intervals, and p values were reported.

For assessment of predicted drug sensitivity, the pRRophetic R package [65] was used with normalized RNA-seq counts from normal, overweight, and obese patient tumors to estimate the half maximal inhibitory concentration (IC50) of 130 drugs from the Cancer Genomic Project (CGP; ref. [66]).

## Statistical analysis

All data analyses were conducted with RStudio and Rx64 3.6.0 software environment. Differences in patient demographic and tumor characteristics between obese, overweight, and normal BMI were analyzed using Fishers exact test. Predicted drug sensitivity was assessed using t-tests to determine differences between obese and overweight BMI patients and normal BMI

patients. Gene expression between colon adenocarcinoma and normal colon tissue in the CRC cohorts was assessed within the Oncomine database using t-tests. A significance level of 0.05 was established for all statistical tests.

## Results

### Patient and tumor characteristics

The TCGA COAD cohort contained both RNA-seq data and weight and height data for 232 patients out of 454 total patients. Weight and height data was used to calculate BMI and classify patients as underweight, normal, overweight, and obese while RNA seq data was used to classify the tumors by the CMS. After exclusion of the one underweight patient, a final cohort of 231 patient's demographic and clinical tumor data was assessed across the CMS categories (S1 Table). Approximately 85% of patient samples in our final cohort were from people located in the US and no significant differences were observed in the geographic site of patient samples across the CMS categories (S1 Table). No significant differences in sex, age, ethnicity, race, tumor stage, or lymph node ratio were observed across the CMS categories. In contrast, tumor location was significantly different across the CMS categories (p = 0.004). The BMI classification of patients was significantly different across the CMS categories (p = 0.040).

The patient's demographic and clinical tumor data was also assessed across the normal, overweight, and obese categories (Table 1). No significant differences in sex, age, or ethnicity were observed across the BMI categories. In contrast, race was significantly different across the BMI categories (p = 0.001): a higher proportion of Asian patients were observed in the normal (10%) compared to the obese (0%) BMI category while a greater proportion of Black or African American patients were observed in the obese (30%) compared to the normal (18%) BMI category. No significant differences in the stage and location of the tumor, nor the lymph node ratio, was observed across the BMI categories. In contrast, the CMS classification of tumors was significantly different across the BMI categories (p = 0.040): a greater proportion of CMS3 tumors was observed in the obese (22%) compared to the normal (4%) BMI category. Consistent with this observation, the highest average BMI was in the CMS3 group (31.2) followed CMS4 (28.6), CMS1 (27.4), and CMS2 (27.1). We also observed in obese patients a significant difference (p = 0.023) in the percentage of Black or African American patients across the BMI categories: CMS3 had the highest percentage (50%) and CMS1 (0%) had the lowest of obese Black or African American patients. Taken together, these preliminary findings suggest that there may be obesity-linked racial differences across the CMS categories.

### CMS specific differentially expressed genes

Our finding that the proportion of CMS3 tumors differed across BMI categories suggested that there may be CMS specific transcriptomic differences in tumors from the obese BMI category compared to the normal and overweight BMI categories. Thus, we examined differentially expressed genes (DEGs) from the RNA seq data between normal, overweight, and obese BMI patients for each CMS. As shown in Fig 2A, Euler diagrams of DEGs demonstrate a unique pattern of overlapping DEGs for each CMS. A significant difference (p < 0.001) in the percentage of obesity-regulated DEGs (overlap between the obese vs. normal and obese vs. overweight comparisons) was observed. CMS1 (70%) and CMS4 (68%) had greater obesity-related DEG overlap than CMS2 (7%) and CMS3 (3%). We next examined the extent to which the obese vs. normal DEGs overlapped between the four CMS categories. The greatest overlap in DEGs was between CMS2 and CMS4 (Fig 2B) where 50 of the obese versus normal DEGs were overlapped (out of 133 and 383 total for CMS2 and CMS4, respectively). A similar result was observed in the overweight vs. normal comparison (Fig 2B). In contrast, we observed in

**Table 1. Patient demographics and tumor characteristics.**

| | Total (n = 231) | | Normal[†] (n = 77) | | Overweight[†] (n = 80) | | Obese[†] (n = 74) | | |
|---|---|---|---|---|---|---|---|---|---|
| | n | % | n | % | n | % | n | % | P-value |
| *Sex** | | | | | | | | | 0.157 |
| Female | 108 | 47 | 37 | 48 | 31 | 39 | 40 | 54 | |
| Male | 123 | 53 | 40 | 52 | 49 | 61 | 34 | 46 | |
| *Age** | | | | | | | | | 0.104 |
| 30–39 | 9 | 4 | 2 | 3 | 6 | 8 | 1 | 1 | |
| 40–49 | 25 | 11 | 11 | 14 | 8 | 10 | 6 | 8 | |
| 50–59 | 45 | 19 | 10 | 13 | 15 | 19 | 20 | 27 | |
| 60–69 | 62 | 27 | 17 | 22 | 19 | 24 | 26 | 35 | |
| 70–79 | 60 | 26 | 23 | 30 | 23 | 29 | 14 | 19 | |
| 80–89 | 28 | 12 | 12 | 16 | 9 | 11 | 7 | 9 | |
| 90> | 2 | 1 | 2 | 3 | 0 | 0 | 0 | 0 | |
| *Ethnicity** | | | | | | | | | 0.205 |
| Hispanic or Latino | 3 | 1 | 1 | 1 | 1 | 1 | 1 | 0 | |
| Not Hispanic or Latino | 221 | 96 | 71 | 92 | 77 | 96 | 73 | 99 | |
| Not Reported | 7 | 3 | 5 | 6 | 2 | 2 | 0 | 0 | |
| *Race** | | | | | | | | | 0.001 |
| American Indian or Alaska Native | 1 | 0 | 0 | 0 | 0 | 0 | 1 | 1 | |
| Asian | 8 | 3 | 8 | 10 | 0 | 0 | 0 | 0 | |
| Black or African American | 51 | 22 | 14 | 18 | 15 | 19 | 22 | 30 | |
| White | 171 | 74 | 55 | 71 | 65 | 81 | 51 | 69 | |
| *Tumor Location** | | | | | | | | | 0.476 |
| Ascending Colon | 41 | 18 | 14 | 18 | 16 | 20 | 11 | 15 | |
| Cecum | 57 | 25 | 18 | 23 | 19 | 24 | 20 | 27 | |
| Descending Colon | 12 | 5 | 4 | 5 | 2 | 2 | 6 | 8 | |
| Hepatic Flexure | 14 | 6 | 7 | 9 | 4 | 5 | 3 | 4 | |
| Rectosigmoid Junction | 1 | 0 | 0 | 0 | 0 | 0 | 1 | 1 | |
| Sigmoid Colon | 62 | 27 | 18 | 23 | 21 | 26 | 23 | 31 | |
| Splenic Flexure | 5 | 2 | 0 | 0 | 4 | 5 | 1 | 1 | |
| Transverse Colon | 23 | 10 | 11 | 14 | 6 | 8 | 6 | 8 | |
| Not Reported | 16 | 7 | 5 | 6 | 8 | 10 | 3 | 4 | |
| *Tumor Stage** | | | | | | | | | 0.547 |
| Stage I | 33 | 14 | 11 | 14 | 10 | 12 | 12 | 16 | |
| Stage II | 94 | 41 | 38 | 49 | 29 | 36 | 27 | 36 | |
| Stage III | 79 | 34 | 20 | 26 | 32 | 40 | 27 | 36 | |
| Stage IV | 25 | 11 | 8 | 19 | 9 | 11 | 8 | 11 | |
| *Lymph Node Ratio** | | | | | | | | | 0.075 |
| LNR0 | 123 | 53 | 41 | 53 | 43 | 54 | 39 | 53 | |
| LNR1 | 57 | 25 | 14 | 18 | 22 | 28 | 21 | 18 | |
| LNR2 | 19 | 8 | 5 | 6 | 8 | 10 | 6 | 8 | |
| LNR3 | 10 | 4 | 3 | 4 | 5 | 6 | 2 | 3 | |
| LNR4 | 11 | 5 | 5 | 6 | 2 | 2 | 4 | 5 | |
| Not Reported | 11 | 5 | 9 | 12 | 0 | 0 | 2 | 3 | |
| *Consensus Molecular Subtype** | | | | | | | | | 0.040 |
| CMS1 | 37 | 16 | 17 | 22 | 11 | 14 | 9 | 12 | |
| CMS2 | 54 | 23 | 20 | 26 | 22 | 28 | 12 | 16 | |

(*Continued*)

**Table 1.** (Continued)

| | Total | | Normal† | | Overweight† | | Obese† | | |
|---|---|---|---|---|---|---|---|---|---|
| | (n = 231) | | (n = 77) | | (n = 80) | | (n = 74) | | |
| | **n** | **%** | **n** | **%** | **n** | **%** | **n** | **%** | **P-value** |
| CMS3 | 31 | 13 | 3 | 4 | 12 | 15 | 16 | 22 | |
| CMS4 | 86 | 37 | 30 | 39 | 26 | 32 | 30 | 32 | |
| Unassigned | 23 | 10 | 7 | 9 | 9 | 11 | 7 | 11 | |

* Significance across score categories by Fishers Exact test
† BMI 19–24.9 (Normal), BMI 25–29.9 (Overweight), BMI ≥30 (Obese).

the obese vs. overweight DEGs that the greatest overlap in DEGs was between CMS3 and CMS4 (Fig 2B), where 31 of the obese versus overweight DEGs were overlapped (out of 322 and 518 total for CMS3 and CMS4, respectively). Volcano plots of DEGs were generated to examine the pattern of overexpressed vs. underexpressed transcripts between normal, overweight, and obese BMI patients for each CMS (S1 Fig). We observed that only CMS3 had more overexpressed transcripts than underexpressed transcripts for the obese vs. normal comparison while both CMS3 and CMS4 had more overexpressed transcripts than underexpressed transcripts for the overweight vs. normal comparison. In contrast, all CMS groups had less overexpressed transcripts than underexpressed transcripts for the obese vs. overweight comparison.

To directly examine the impact of obesity across CMSs, we used an interaction term for obesity:CMS in the DESeq2 linear model followed by pairwise comparisons between CMSs. The Euler diagrams of DEGs demonstrate a unique pattern of overlapping DEGs for each CMS (S2 Fig). The greatest overlap in DEGs across CMS comparisons was observed from CMS1 (18 genes) while the least overlap was observed in CMS4 (1 gene) (S2 Fig).

To examine the clinical relevance of the CMS specific DEGs, we generated Kaplan-Meier plots (59) to assess overall survival and relapse-free survival in CRC cohorts from the United States. Using the top 20 upregulated DEGs identified in the obese to normal BMI comparison, we observed significantly reduced overall survival, adjusted for age, gender, stage and grade, for high expression of the CMS3 DEGs (p = 0.016; HR = 3.73, 95% CI 1.27–10.93) and CMS4 DEGs (p = 0.026; HR = 2.79, 95% CI 1.13–6.88) (Fig 3). Significantly reduced relapse-free survival, adjusted for age, gender, stage and grade, was observed for high expression of CMS2 DEGs (p = 0.016; HR = 13.47, 95% CI 1.61–113) and CMS3 DEGs (p = 0.042; HR = 2.48, 95% CI 1.03–5.96) (Fig 2B). Only high expression of CMS1 DEGs was associated with significantly reduced relapse-free survival in the obese to overweight BMI comparison (p = 0.016; HR = 14.33, 95% CI 1.23–167). Using the top 20 upregulated DEGs identified in the overweight to normal BMI comparison, we also observed significantly reduced overall and relapse-free survival, adjusted for age, gender, stage and grade, for high expression of CMS2, CMS3, and CMS4 DEGs (S3 Fig). Taken together, these preliminary findings indicate CMS specific differences in obesity-regulated DEGs.

## CMS specific gene set enrichment

To further examine obesity-related transcriptomic differences in the four CMSs, we performed a gene set enrichment analysis comparing RNA seq data from obese to normal BMI patients for each CMS. As shown in Fig 4, immune-related Hallmark gene sets with an FDR (p < 0.05) were enriched in CMS1, CMS2, and CMS4. Immune-related gene sets were almost exclusively

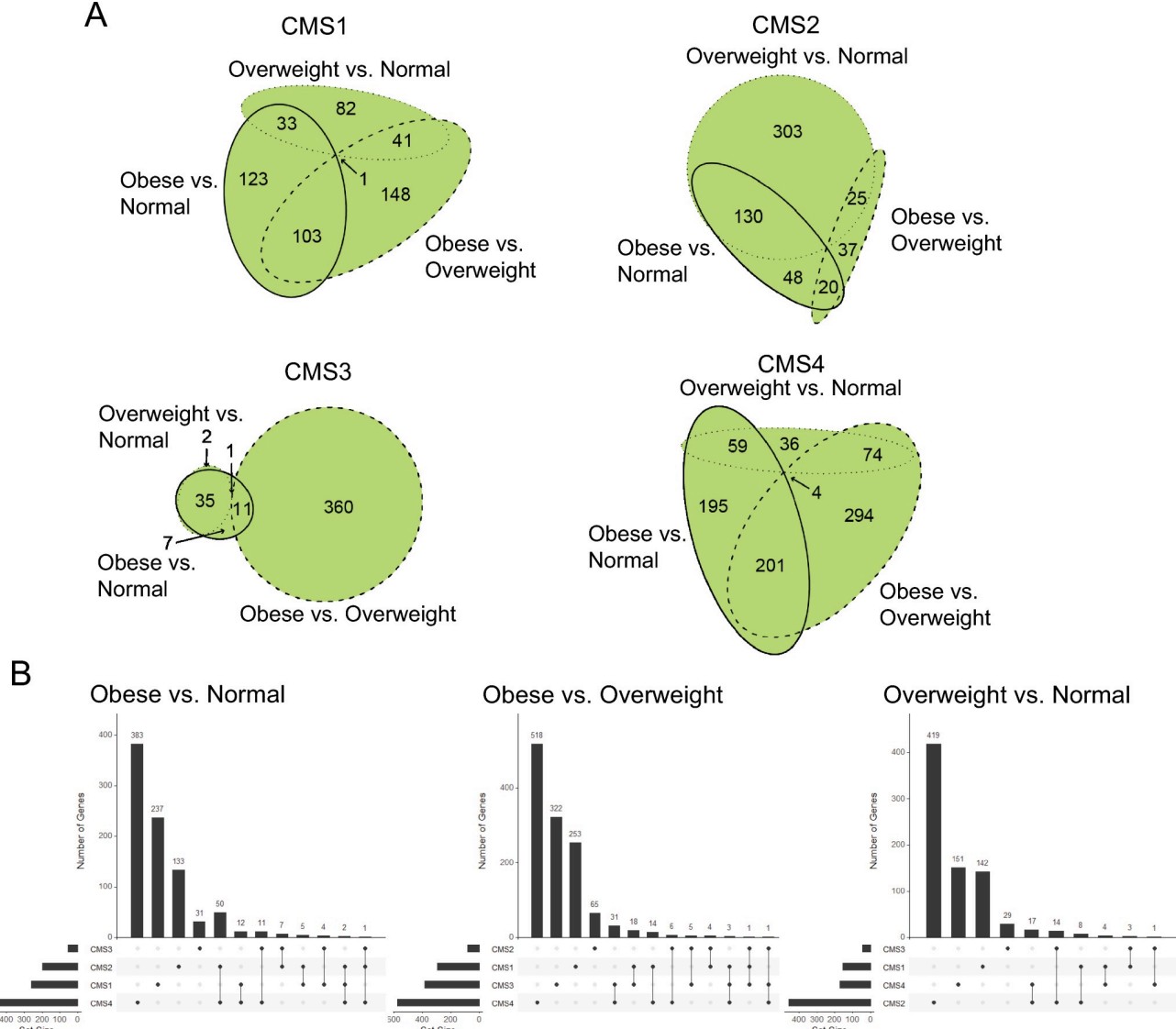

**Fig 2. Differential expressed gene analysis reveals CMS-specific differences between BMI groups.** (A) Euler diagrams were used to visualize the weighted overlap of DESeq2-obtained DEGs (MeanBase > 10, FDR p value < 0.05) between Obese vs. Normal, Obese vs. Overweight, and Overweight vs Normal DEGs for each CMS category. The R package eulerr was used to construct the Euler diagrams. (B) Upset plots were used to visualize the intersection of DEGs between the four CMS categories for each BMI comparison. The R package UpsetR was used to construct the Upset plots.

enriched in both CMS1 (8 out of 10) and CMS2 (2 out of 2). In contrast, Cell cycle- and Metabolism-related Hallmark gene sets were highly enriched in CMS4 tumors. Interestingly, significant Hallmark gene set enrichment was not observed in CMS3. Analysis of Hallmark gene set enrichment with an FDR (p < 0.05) in the obese vs. overweight comparison revealed that CMS1 gene set enrichment was dominated by immune-related gene sets (8 out of 14) (Fig 4). In contrast, immune-related gene sets were not enriched in CMS2, CMS3, and CMS4 except for the coagulation gene set in CMS3. The only significant enrichment in CMS2 was the Hedgehog signaling gene set. Unique to CMS4 were 18 enriched gene sets related to proliferation (MTORC signaling), metabolic (cholesterol homeostasis and glycolysis), and stromal signaling (WNT beta catenin signaling, TGF beta signaling, and Notch signaling). Although, the WNT beta catenin signaling gene set overlapped between CMS3 and CMS4.

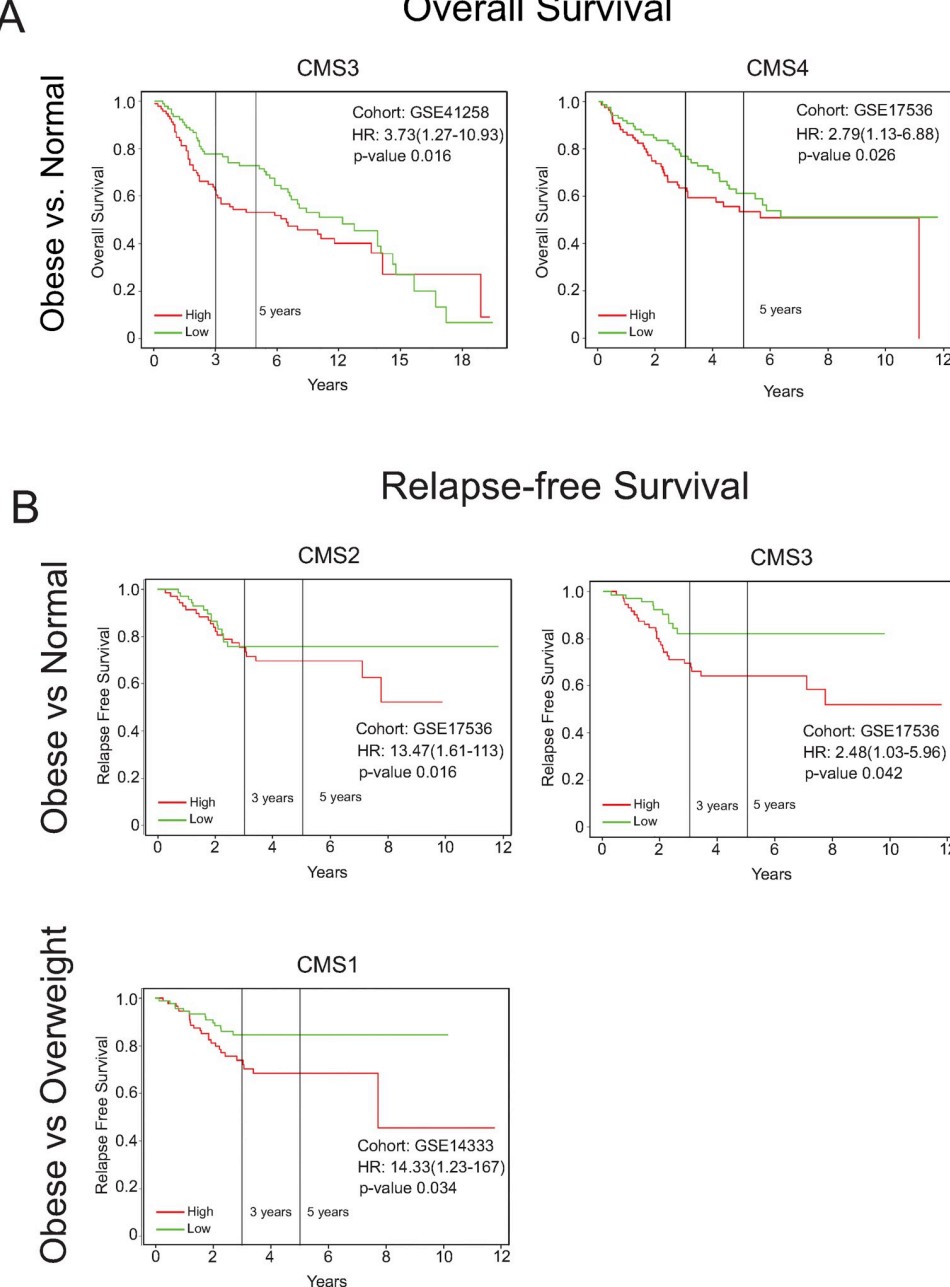

**Fig 3. Prognostic patient outcomes reveal CMS-specific differences between obese BMI groups.** The expression of the top 20 significantly upregulated DESeq2-obtained DEGs (MeanBase > 10, FDR p value < 0.05) in the obese vs. normal and obese vs. overweight comparisons for each CMS category were assessed in GSE17536 and GSE41258 for overall survival (A) and GSE14333 and GSE17536 for relapse-free survival (B) using the PROGgeneV2 tool. Survival analyses were adjusted for age, stage, and gender covariates and bifurcated based on median expression. The hazard ratios, 95% confidence intervals, and p values were reported for the Kaplan-Meier plots.

A sensitivity analysis was performed to determine whether inclusion of Asian patients, which were all categorized as normal BMI, affected gene set enrichment analysis comparing RNA seq data from obese to normal and overweight to normal BMI patients for each CMS. As shown in S4B Fig, the enrichment of Hallmark gene sets in obese patients mirrored the findings observed with whole patient population (Fig 4A). All 27 enriched gene sets in the whole

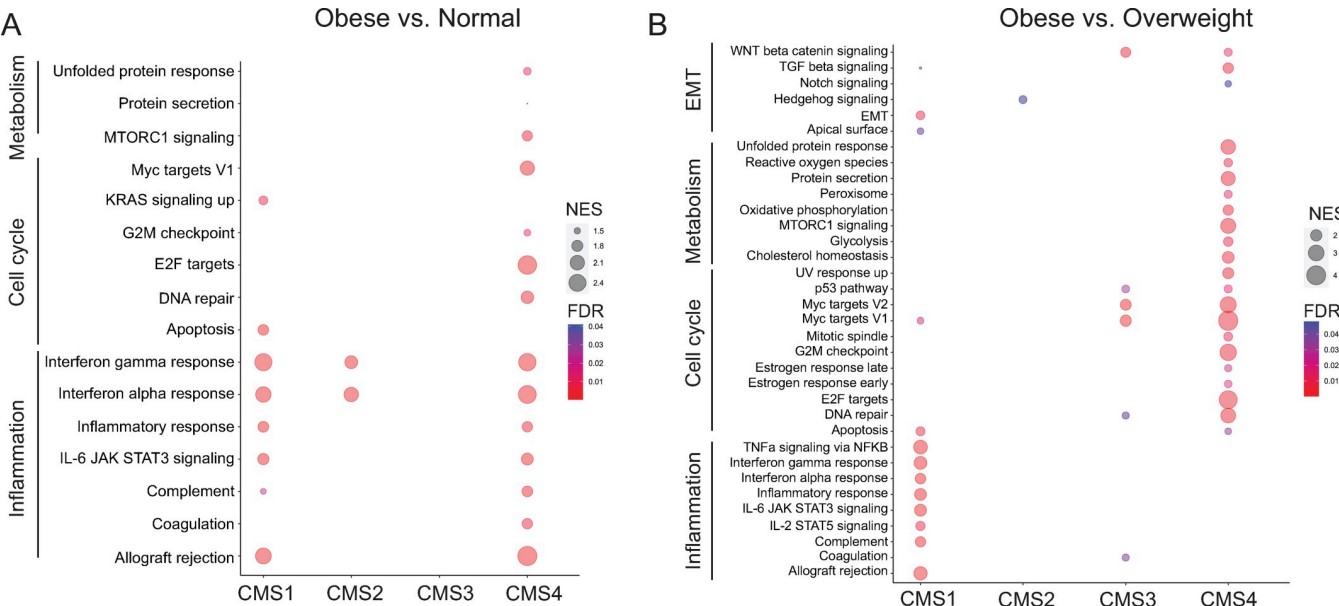

**Fig 4. Gene set enrichment of Hallmark gene sets reveal CMS-specific differences between obese BMI groups.** Normalized RNA-seq counts obtained from DESeq2 were used for Gene Set Enrichment Analysis (GSEA). Hallmark gene sets were assessed in the obese vs. normal (A) and obese vs. overweight (B) comparisons for each CMS category. Bubbles for gene sets with a false discovery rate q-value less than 0.05 were reported. The size of bubbles represents the normalized enrichment score (NES). The color of bubbles represents false discovery rate q-value (FDR).

population were enriched in the sensitivity analysis. Further, the normalized enrichment score was highly correlated ($\beta = 1.001$, SE = 0.108, p = 5.0 x $10^{-9}$) between the whole patient population and the population excluding Asians. However, the exclusion of Asians in the sensitivity analysis did result in significant enrichment of three EMT gene sets in obese CMS4 tumors that did not reach significance in the whole patient population. Taken together, these preliminary findings indicate CMS specific differences in obesity-regulated immune, metabolic, and stromal signaling gene set enrichment.

In addition to examining obesity-related transcriptomic differences, we examined transcriptomic differences between overweight vs. normal BMI patients for each CMS (S5A Fig). Hallmark gene set enrichment with an FDR (p < 0.05) revealed immune-related gene set enrichment in CMS2 and CMS4. In contrast, gene set enrichment in CMS1 was strongly related to metabolic processes (heme metabolism, fatty acid metabolism, and bile acid metabolism). The only significant enrichment in CMS3 was a metabolic process gene set (oxidative phosphorylation). In a sensitivity analysis for the overweight to normal BMI comparison, we observed that all 21 enriched gene sets from the whole patient population were also significantly enriched when the population excluded Asians (S5B Fig). However, there were an additional 33 significantly enriched gene sets primary in Metabolism related gene sets (20 newly enriched) and Cell Cycle related gene set (9 newly enriched). In addition, 18 of the significantly enriched gene sets were observed in CMS3 tumors. These preliminary findings indicate that racial/ethnic differences may strongly influence transcriptomic differences in tumors from overweight patients.

## CMS specific hub genes

To gain insight into obesity-regulated hub genes within each CMS group, we first constructed a Protein-Protein Interaction (PPI) network using the STRING database module in Cytoscape.

From the PPI network, hub genes were identified using the MCC algorithm of the CytoHubba module in Cytoscape. The top ten highest scoring genes in the obese to normal BMI patients and the obese to overweight comparisons for each CMS are shown in Fig 5 and S2 Table. A sensitivity analysis of the MCC algorithm hub genes was performing using four other topographical algorithms. Hub genes were commonly identified in at least 4 out of the 5 topographical algorithms (S3 Table). We observed that in CMS1 obese to normal BMI comparison there were four hub genes: *Bassoon presynaptic cytomatrix protein* (*BSN*), *Major synaptic vesicle protein p38* (*SYP*), and *RAB3C, member RAS oncogene family* (*RAB3C*), and *Unc-13 homolog A* (*UNC13A*). In contrast, an immune hub containing *interleukin 10* (*IL-10*), *C-C motif chemokine receptor 2* (*CCR2*), and *C-C motif chemokine ligand 13* (*CCL13*) was observed in the obese to overweight BMI comparison for CMS1. Weak interconnectivity was observed in the obese to normal BMI comparison for CMS2, while a hub containing *NK2 homeobox 1* (*NKX2-1*) and *SRY-box transcription factor 2* (*SOX2*) was observed in the obese to overweight BMI comparison for CMS2. A hub containing four Melanoma-associated antigen genes (*MAGEA6*, *MAGEA3*, *MAGEA11*, and *MAGEA12*) was observed in the obese to normal BMI comparison for CMS3, while an interconnected hub network of genes including a somatostatin receptor gene (*SSTR5*) but also a C-C motif chemokine receptor gene (*CCR2*) was observed in the obese to overweight BMI comparison for CMS3. In the obese to normal comparison for CMS4 we observed a hub with UDP glucuronosyltransferase 1 family, polypeptide genes (*UGT1A1* and *UGT1A8*), while a hub network of genes including *neuromedin U receptor 2* (*NMUR2*), *peptide YY* (*PYY*) and *pro-platelet basic protein* (*PPBP*) was observed in the obese to overweight BMI comparison for CMS4. The MCC algorithm hub genes for the overweight to normal BMI comparison were also identified (S6 Fig and S2 Table). Hub genes commonly identified in at least 4 out of the 5 topographical algorithms are shown in S3 Table. In CMS1 there was an overlap in *SYP* with the hub observed in the obese comparison to normal BMI while no overlap in hub genes was observed for CMS2. An overlap in the Melanoma-associated antigen genes (*MAGEA6*, *MAGEA3*, and *MAGEA12*) was observed with the hub observed in the CMS3 obese comparison to normal BMI, while no overlaps were observed in CMS4.

## Obesity specific hub genes across CMS categories

To directly examine the impact of obesity across CMSs, we used the Likelihood ratio test in DESeq2 to identify the obesity effect across CMSs. We observed 1579 obesity-linked DEGs (p adjusted < 0.05; BaseMean > 10; >1 log2FoldChange). We constructed a PPI network from the DEGs using the STRING database and identified hub genes using the MCC algorithm and confirmed in at least 3 other topographical algorithms in CytoHubba as described above. An inflammation gene hub was identified. Hub genes commonly identified were *Fc region receptor III-A (FCGR3A)*, *IL10*, *C-X-C Motif Chemokine Ligand 8* (*CXCL8*), *Integrin Subunit Alpha M* (*ITGAM)*, *Cluster of Differentiation 86 (CD86)*, *Protein Tyrosine Phosphatase Receptor Type C* (*PTPRC)*, and *C-C Motif Chemokine Receptor 5* (*CCR5)* (S7 Fig). This gene set represents a general effect of obesity on the CRC transcriptome that is not specific for any one CMS.

To examine the CRC relevance of the hub genes that overlapped in more than one CMS obese BMI comparisons, we used transcriptomic microarray data compiled from the NCBI GEO public database using the GENT2 program for an aggregate data analysis and in three independent colon cancer cohorts using the Oncomine database [53, 54] for an analysis of cohort variability. We observed that *PYY* transcript expression was significantly (p < 0.001) downregulated (Log2 fold change = -4.115) in aggregate colon cancer data (Fig 6A) and in all three colon cancer cohorts (S8A Fig). In contrast, *PPBP* transcript expression was significantly (p < 0.001) upregulated (Log2 fold change = 2.803) in aggregate colon cancer data (Fig 6B)

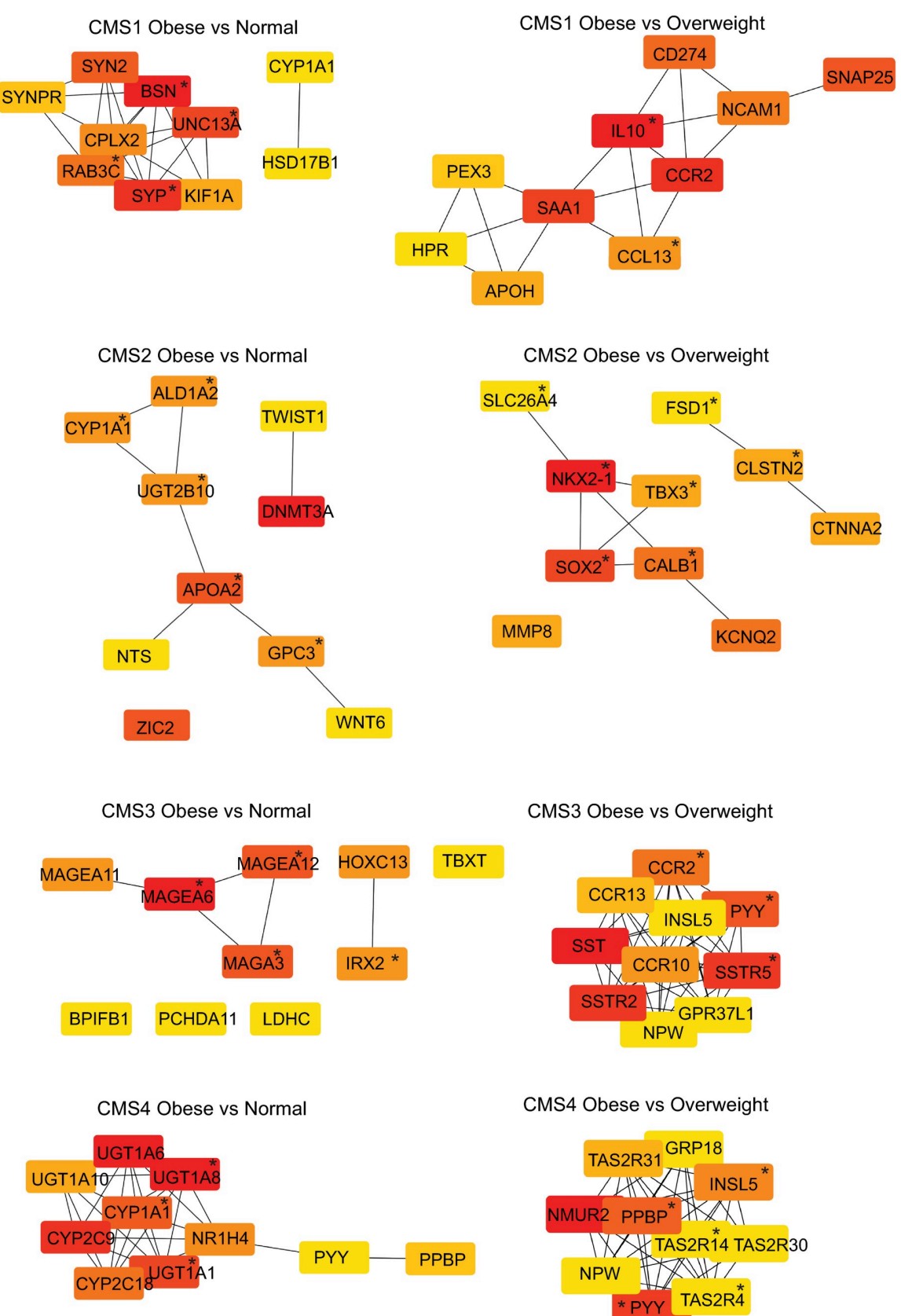

**Fig 5. Hub gene analysis reveal CMS-specific differences between obese BMI groups.** DESeq2-obtained DEGs (MeanBase > 10, FDR p value < 0.05) were used to construct a protein-protein interaction (PPI) network from the STRING database in Cytoscape. The Cytohubba package in Cytoscape was used to perform the hub gene analysis from the PPI network for the obese vs. normal (left panels) and obese vs. overweight (right panels) comparisons for each CMS category. Hub genes were identified using the maximal clique centrality (MCC) method to obtain the top 10 ranked genes in all modules. The intensity and color (high, red; orange, medium; yellow, low) of the hub genes is shown.

and in two of the cohorts (S8B Fig). *INSL5* was significantly (p < 0.001) downregulated (Log2 fold change = -3.966) in colon cancer (Fig 6C) and in one of the three colon cancer cohorts (S8C Fig). Differential regulation of the *NPW* transcript expression was observed in aggregate colon cancer data (Fig 6D) and in one of the three cohorts (S6D Fig). Finally, *CCR2* gene expression was significantly (p < 0.001) downregulated (Log2 fold change = -0.706) in colon cancer (Fig 6E). Taken together our analysis of CMS-specific hub genes infers that there may exist not yet considered mechanisms playing a role in modulating the colon cancer tumor transcriptome. The CMS-specific hub gene preliminary findings are useful for hypothesis testing to examine new colon cancer mechanisms and identify new therapeutic targets to treat colon cancer.

## CMS specific predicted drug sensitivity

We next examined whether obesity modulated the predicted drug sensitivity of 130 drugs from the Genomics Drug Sensitivity in Cancer (GDSC) screen in a similar manner across the four CMSs. We used a phenotype prediction method in which cell line drug response is applied to patient transcriptomic data [67]. The front line chemotherapy drugs fluoropyrimidine, irinotecan, and capecitabine [68] were not available for analysis in our 130 drug set but data for 8 other drugs targeting DNA replication were available and as was data for another 122 drugs in pathways that have been targeted for CRC (apoptosis, cell cycle, chromatin histone acetylation, cytoskeleton, epidermal growth factor receptor (EGFR), extracellular signal regulated kinase (ERK)/mitogen activated protein kinase (MAPK), genome integrity, insulin-like growth factor receptor (IGFR), c-Jun N-terminal kinase (JNK), metabolism, mitosis, other kinases, phosphoinositide 3 kinase (PI3K)/mammalian target of rapamycin (MTOR), protein stability, receptor tyrosine kinase (RTK), and WNT signaling) [69–71]. We observed significantly (p < 0.05) increased predicted drug sensitivity for 32 and reduced predicted drug sensitivity for 4 of the 130 drugs between obese and normal BMI patient tumors, primarily in CMS1 (22 drugs) but also in CMS4 (10 drugs) and CMS2 (3 drugs) but not CMS3 (1 drug) (S4 Table). There were no significant differences in predicted drug sensitivity between obese and normal BMI patient tumors in any of the CMS categories for drugs targeting EGFR, ERK/MAPK, IGF1R, and RTK (S4 Table). In contrast, we observed significantly (p < 0.05) increased predicted drug sensitivity for 4 drugs targeting DNA replication (Fig 7A and S4 Table). Further, significantly (p < 0.05) increased predicted drug sensitivity for 4 drugs targeting MTOR (Fig 7B and S4 Table), including temsirolimus in both CMS1 and CMS4, was observed. Significantly (p < 0.05) reduced predicted drug sensitivity for 2 drugs targeting metabolism (peroxisome proliferator-activated receptor (PPAR)) in both CMS2 and CMS4 was also observed (Fig 7C and S4 Table). Taken together our preliminary findings indicate that there are CMS-specific differences in predicted drug sensitivity between obese and normal BMI patient tumors.

We next examined predicted drug sensitivity of 130 drugs from the GDSC screen in a similar manner across the four CMSs for tumors from overweight compared to normal BMI patients. We observed significantly (p < 0.05) increased predicted drug sensitivity for 14 and reduced predicted drug sensitivity for 7 of the 130 drugs primarily in CMS2 (17 drugs) but

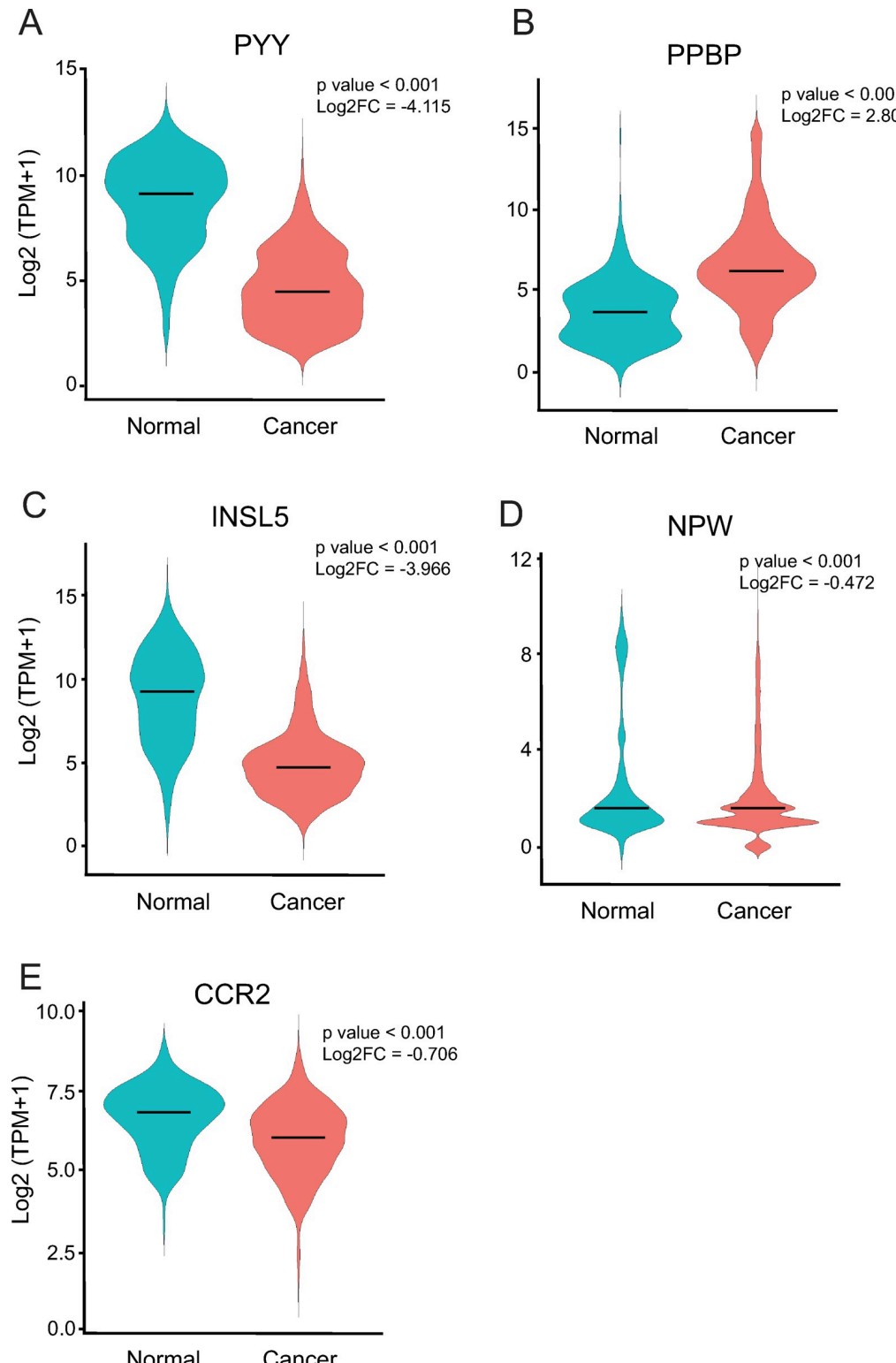

**Fig 6. Hub gene expression is relevant to colon cancer.** Hub genes were queried for mRNA expression using Affymetrix U133Plus2 platform microarray data from normal colon tissue (Normal) and colon cancer (Cancer) in the GEO database and the GENT2 program. Violin plots were generated using ggplot2 in R studio. The median is represented as a horizontal bar in the violin sample plot. The p value from t-tests and the Log2 fold change (Log2FC) are shown. (A) PYY, (B) PPBP, (C) INSL5, (D) NPW, and (E) CCR2.

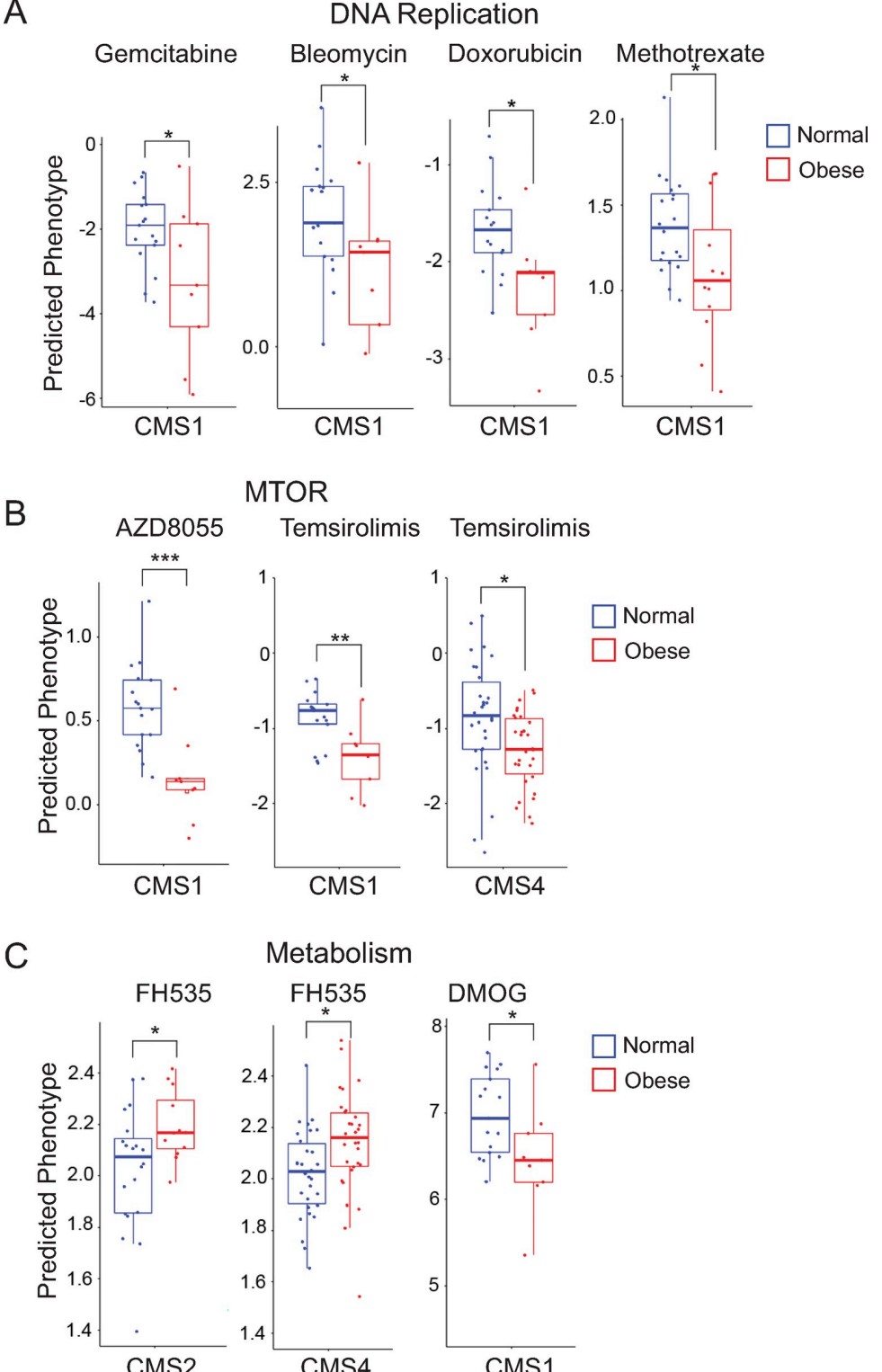

**Fig 7. Predicted drug sensitivity analysis reveal CMS-specific differences between obese BMI groups.** Estimated half maximal inhibitory concentration (IC50) of drugs targeting DNA Replication (A), MTOR (B), and Metabolism (C) for tumors from normal (blue) and obese (red) BMI patients across the CMS categories is shown. To assess predicted drug sensitivity, the pRRophetic R package was used with normalized RNA-seq counts. Box plots with p values from t-tests are shown.

also in CMS3 (3 drugs) and CMS1 (1 drug) but not CMS4 (0 drugs) (S5 Table). Consistent with the findings in the obese to normal comparison, there was increase predicted drug sensitivity for the drugs targeting DNA replication (including methotrexate) and reduced predicted drug sensitivity for the drug FH535 which targets PPAR (S5 Table). Further we observed in CMS2 tumors differential predicted drug sensitivity for two CMS2 canonical pathways: significantly increased predicted drug sensitivity for the drugs targeting EGFR and significantly reduced predicted drug sensitivity for a drug targeting WNT signaling were observed S5 Table). Taken together, these findings suggest that there is a potential for obesity-based precision therapeutics. However, our preliminary findings will require validation with both *in vitro* drug testing using cell models of obesity-linked inflammation and insulin resistance and drug studies in obese animal models using CMS-specific patient-derived xenografts prior to clinical evaluation.

## Discussion

In the current study, we examined the transcriptomic profile of tumors from obese patients compared to healthy and overweight BMI patients. We found that obesity differentially affected cancer pathways, hub genes, prognostic patient survival, and predicted drug sensitivity in a molecular subtype specific manner. Our findings are consistent with mechanistic findings indicating that there are multiple mechanisms linking obesity and colon cancer [17, 18, 40]. Our findings are significant because they suggest that the CMS of the tumor is an important factor in the link between obesity and colon cancer.

CMS1 tumors are categorized as the immune subtype based on the strong infiltration of immune cells and microsatellite instability (MSI) [5, 6]. Our findings that immune-related Hallmark gene sets were enriched in CMS1 tumors in all comparisons with obese patients suggests that obesity enhances the immune phenotype of this subtype. In support of the GSEA findings, we observed that an immune gene hub was identified in CMS1 tumors from obese compared to overweight BMI patients. These findings are consistent with the pathophysiological state of obesity being associated with systemic inflammation [15, 16]. Further, we observed immune-related Hallmark gene set enrichment in CMS2 and CMS4 tumors in the obese to normal patient BMI comparison but not in the obese to overweight patient BMI comparison. Consistent with this finding, immune-related Hallmark gene set enrichment in CMS2 and CMS4 tumors was observed in the overweight to normal patient BMI comparison. These findings suggest a differential role of inflammation in the subtypes of colon cancer tumors based on whether the patient is overweight or obese. The commonality of immune-related Hallmark gene set enrichment in 3 of the 4 subtypes and our finding of an inflammation-linked gene hub across CMSs are important because an inflammatory risk score is an independent predictor for stage II colon cancer prognosis [72] and a circulating inflammation signature is a strong prognostic factor of progression-free and overall survival of patients with metastatic CRC [31]. Consistent with these findings, a higher dietary inflammatory potential is associated with higher CRC risk [73].

The deregulation of cellular energetics resulting in the reprogramming of energy metabolism plays a role in tumorigenesis [74] and has been identified as an emerging hallmark of cancer [75]. Our observation that CMS2, CMS3 and CMS4 enrichment in the Myc gene sets Myc targets V1 and Myc targets V2 suggests that key cell signaling and metabolic pathways within tumor cells driving tumor growth and progression are differentially regulated by obesity. It has been hypothesized that obesity-derived factors (e.g. circulating hormones, adipokines, inflammatory cytokines, and dietary factors) converge on these key cell signaling and metabolic pathways [40]. The enrichment of metabolism related gene sets was concentrated in CMS4 tumors.

Consistent with this finding, we also observed that an obesity-linked network of hub genes related to the *UGT1A* gene locus was downregulated in CMS4 tumors. In agreement with this observation, we found that *UGT1A1*, *UGT1A6*, and *UGT1A8* expression was reduced in colon adenocarcinoma compared to normal colon tissue in 3 separate large CRC cohorts. The UGT1 subfamily of enzymes reduces the biological activity and enhances the solubility of lipophilic substrates through the process of glucuronidation [76]. UGT activity has been hypothesized to modulate energy metabolism by altering cellular pools UDP-sugars which are glycolytic intermediates or through interaction with pyruvate kinase (PKM2), a glycolytic enzyme [76, 77].

Enrichment of EMT-related Hallmark gene sets and metabolism-related gene sets was observed in CMS4 tumors in the obese to overweight patient BMI comparison. These findings suggest that obesity enhances the mesenchymal features CMS4 subtype and induces a metabolic phenotype which is typically associated with CMS3 tumors [5, 6]. It has been proposed that a metabolic shift in the canonical CMS2 tumors possibly due to KRAS mutations and copy number events results in CMS3 tumors whereas the stromal-enriched inflamed tumor microenvironment is the driver for the development of CMS4 tumors from the CMS2 subtype [5]. Interestingly, we observed an obesity-induced enrichment of EMT- and metabolism-related Hallmark gene sets in CMS4 tumors and that a greater proportion of CMS3 tumors in obese compared to normal BMI patients. Our later finding is consistent with a report [78] that patients with CMS3 tumors in a Stage II-IV CRC cohort are more likely (OR 3.5, 95% CI 1.1–11.4) to have type 2 diabetes, an obesity-linked disease. Whether obesity plays a role in shifting CMS2 tumors to CMS3 or CMS4 is not known.

A hallmark feature of the mesenchymal CMS4 subtype is complement activation [5, 6] with a platelet signature [79]. Indeed, we observed that complement activation was an obesity-enriched gene set in CMS4 tumors and that the platelet marker PPBP was found to be an obesity-linked hub gene in CMS4 tumors. We also observed complement activation enrichment and *PPBP* as a hub gene in CMS2 tumors in the overweight to normal comparison, suggesting a platelet signature in CMS2 tumors from overweight patients. Platelets help initiate and coordinate the immune response including resolution of inflammation [80, 81]. Visceral obesity is associated with persistent platelet activation [82]. It has been observed that platelet to lymphocyte ratio (PLR) is higher in CRC patients with the Metabolic syndrome and that PRL is associated with poorer overall survival [83]. Further, it has recently been reported that microparticles released from thrombin-activated platelets obtained from obese women induce the expression of EMT and EndMT marker genes when incubated with human colon (HT29) cancer cells [84], suggesting that activated platelets can modulate colon cancer progression.

We found that obese patients with CMS1 tumors had worse prognostic relapse-free survival compared to overweight patients. Our finding is consistent the previously reported poor survival rate after relapse in patients with CMS1 tumors [6], and the reported stronger association between BMI and MSI-high CRC and microsatellite-stable CRC [18]. We also observed that obese patients with CMS4 tumors had worse prognostic overall survival compared to normal BMI patients. Our findings with CMS4 tumors are consistent with the worse patient overall survival has been reported for CRC patients with CMS4 tumors [6] and a report that a fibroblast-like and elevated myeloid signature is correlated with poor patient survival [85]. Interestingly, we observed that prognostic overall and relapse-free survival was worse in obese patients with CMS3 tumors while relapse-free survival was also worse in obese patients with CMS2 tumors. Taken together our findings suggest that obesity may contribute to worse survival beyond that previously observed in patients with CMS1 and CMS4 tumors. Our findings of worse prognostic survival in patients with CMS4 tumors were not restricted to obese patients; we observed that overall and relapse-free survival was worse in overweight patients too. There is strong epidemiological evidence that obesity is associated with CRC risk [86]. Yet, it has

been reported that an obesity paradox exists for CRC where being overweight is associated with improved survival [87]. However, a recent meta-analysis has reported that CRC recurrence is increased by 33% in overweight compared to normal BMI (p < 0.001) [88]. Indeed, it has been suggested that methodological problems in studies using BMI may affect findings and interpretation of those findings [86].

Counter intuitively, we observed that obesity was associated primarily with increased predicted sensitivity with GDSC drugs including those targeting DNA replication and MTOR, but not metabolism pathways which indicates that not all CRC pathways were equally affected. Increased predicted sensitivity was concentrated in the CMS1, the MSI immune subtype. Our observation that inflammation-related gene sets are enriched in CMS1 tumors from obese patients is consistent with the finding that immune infiltration predicts fluoropyrimidine, a DNA replication-based chemotherapy [89]. Our findings of differential CMS-dependent differences in predicted drug sensitivity are consistent with the observations that differential response to irinotecan-based compared to oxaliplatin-based chemotherapy [90] and chemotherapy plus bevacizumab compared cetuximab [7, 91] in metastatic CRC clinical trials are CMS-dependent. Even though it has been suggested that the current CMS classification does not provide a rationale for targeted therapy in metastatic CRC [92], our findings suggest that obesity-mediated changes in tumor biological pathways may inform drug discovery and rational combination therapies. In contrast to our finding on predicted drug sensitivity in the obese to normal comparison, differential predicted drug sensitivity in the overweight to normal comparison was concentrated in CMS2, the canonical pathways subtype. Consistent with the observation, we observed significantly different predicted drug sensitivity in EGFR and WNT signaling, two CRC canonical pathways.

A limitation of the current study is that it was only performed in the TCGA-COAD cohort which lacked racial diversity. The impact of race/ethnicity may have led to possible confounding. CMS categorization of CRC patients across the BMI categories led to small sample sizes which lowered statistical power for some of the comparisons particularly those with normal BMI patients with CMS3 tumors. Confirmation of our findings will require assembly of a large CRC cohort that contains both transcriptomic and body weight and height data. Additional limitations of the current study are that: 1) confounding factors that are risk factors for CRC such as smoking and alcohol consumption were not assessed; 2) weight loss prior to diagnosis which has been reported in all four colon cancer stages [93, 94], was not assessed; and 3) our findings in CMS1 tumors should be interpreted with caution because CMS1 is associated with familial disease and a younger patient population. Finally, it should be noted that BMI is a proxy for adiposity but may not account for the metabolic health of the patients.

## Conclusions

Our findings support that obesity impacts the CRC tumor transcriptome in a CMS-specific manner. This observation is based not only on CMS-specific associations of obesity in gene set enrichment but also on findings of obesity-related DEGs and the identification of unique hub genes for each CMS in tumors from obese patients. Prognostic patient survival analysis and predicted drug sensitivity support our findings that obesity has CMS-specific associations in colon cancer. Taken together, our findings are consistent with the hypothesis that the obesity-cancer link is mediated by obesity-derived factors which converge on key cell signaling and metabolic pathways; yet this occurs in a CMS-specific manner in colon cancer. These findings will require validation using in vitro and animal models to examine the CMS-dependence of the genes and pathways. Once validated the obesity-linked genes and pathways may represent new therapeutic targets to treat colon cancer in a CMS-dependent manner.

## Supporting information

**S1 Table. Patient demographics and tumor characteristics by CMS category.**
(DOCX)

**S2 Table. Hub gene analysis for the obese vs.** normal, obese vs. overweight, and overweight vs. normal comparisons for each CMS category.
(DOCX)

**S3 Table. Sensitivity analysis to assess maximal clique centrality (MCC) identified hub genes in four additional topographical algorithms.**
(DOCX)

**S4 Table. Predicted drug sensitivity for normal compared to obese BMI categories.**
(DOCX)

**S5 Table. Predicted drug sensitivity for normal compared to overweight BMI categories.**
(DOCX)

**S1 Fig. Differential expressed gene analysis reveals CMS-specific differences between BMI groups.** (A) Volcano plots were used to visualize DESeq2-obtained DEGs (MeanBase > 10, FDR p value < 0.05) between Obese vs. Normal (A), Overweight vs Normal (B), and Obese vs. Overweight (C) comparisons for each CMS category. The R package EnhancedVolcano was used to construct the plots. The ratio of overexpressed to underexpressed DEGs is shown for each volcano plot. The DEGs with a false discovery rate less than 0.05 are shown as red dots while nonsignificant DEGs are represented as green dots. Select highly significant and differentially expressed genes are identified in the plots.
(PDF)

**S2 Fig. Differential expressed gene analysis reveals obesity-linked difference across the CMS categories.** Euler diagrams were used to visualize the weighted overlap of DESeq2-obtained Obese vs. Normal DEGs (MeanBase > 10, FDR p value < 0.05) using an interaction term for obesity:CMS in the DESeq2 linear model for each CMS category. The R package eulerr was used to construct the Euler diagrams.
(PDF)

**S3 Fig. Prognostic patient outcomes reveal CMS-specific differences between overweight and normal BMI groups.** The expression of the top 20 significantly upregulated DESeq2-obtained DEGs (MeanBase > 10, FDR p value < 0.05) in the overweight vs. normal comparisons for each CMS category were assessed in GSE17536 and GSE41258 for overall survival (A) and GSE14333 and GSE17536 for relapse-free survival (B) using the PROGgeneV2 tool. Survival analyses were adjusted for age, stage, and gender covariates and bifurcated based on median expression. The hazard ratios, 95% confidence intervals, and p values were reported for the Kaplan-Meier plots.
(PDF)

**S4 Fig. Gene set enrichment of Hallmark gene sets reveal CMS-specific differences between obese and normal BMI groups in the whole population and the population without the Asian patients.** Normalized RNA-seq counts obtained from DESeq2 were used for Gene Set Enrichment Analysis (GSEA). Hallmark gene sets were assessed in the obese vs. normal comparisons in the whole population (A) and the population without the Asian patients (B) for each CMS category. Bubbles for gene sets with a false discovery rate q-value less than 0.05 were reported. The size of bubbles represents the normalized enrichment score (NES). The color of

bubbles represents false discovery rate q-value (FDR).
(PDF)

**S5 Fig. Gene set enrichment of Hallmark gene sets reveal CMS-specific differences between overweight and normal BMI groups in the whole population and the population without the Asian patients.** Normalized RNA-seq counts obtained from DESeq2 were used for Gene Set Enrichment Analysis (GSEA). Hallmark gene sets were assessed in the overweight vs. normal comparison in the whole population (A) and the population without the Asian patients (B) for each CMS category. Bubbles for gene sets with a false discovery rate q-value less than 0.05 were reported. The size of bubbles represents the normalized enrichment score (NES). The color of bubbles represents false discovery rate q- value (FDR).
(PDF)

**S6 Fig. Hub gene analysis reveal CMS-specific differences between overweight and normal BMI groups.** DESeq2-obtained DEGs (MeanBase > 10, FDR p value < 0.05) were used to construct a protein-protein interaction (PPI) network from the STRING database in Cytoscape. The Cytohubba package in Cytoscape was used to perform the hub gene analysis from the PPI network for the overweight vs. normal comparison for each CMS category. Hub genes were identified using the maximal clique centrality (MCC) topological algorithm to obtain the top 10 ranked genes in all modules. Hub genes identified in at least three of the four topological algorithms are designated with an asterisk. The intensity and color (high, red; orange, medium, yellow, low) of the hub genes is shown.
(PDF)

**S7 Fig. CMS-independent hub gene analysis reveal obesity-specific differences between obese and normal BMI groups.** Obesity-linked DEGs (MeanBase > 10, FDR p value < 0.05) obtained using the Likelihood ratio test in DESeq2 were used to construct a protein-protein interaction (PPI) network from the STRING database in Cytoscape. The Cytohubba package in Cytoscape was used to perform the hub gene analysis from the PPI network for the obese vs. normal comparison. Hub genes were identified using the maximal clique centrality (MCC) topological algorithm to obtain the top 10 ranked genes in all modules. Hub genes identified in at least three of the four topological algorithms are designated with an asterisk. The intensity and color (high, red; orange, medium, yellow, low) of the hub genes is shown.
(PDF)

**S8 Fig. Hub gene expression is relevant to colon adenocarcinoma in independent cancer patient cohorts.** Hub genes were queried for mRNA expression using the Oncomine database. Hub gene expression in colon adenocarcinoma (Carcinoma) versus normal patient samples from the Hong Colorectal (Normal, n = 12; Carcinoma, n = 70), Skrzypczak Colorectal (Normal, n = 24; Carcinoma, n = 36), and Kaiser Colorectal (Normal, n = 5; Carcinoma, n = 41) cohorts. The p value from t-tests are shown. (A) *PYY*, (B) *PPBP*, (C) *INSL5*, and (D) *NPW*.
(PDF)

## Acknowledgments

The results shown here are in whole or part based upon data generated by the TCGA Research Network: https://www.cancer.gov/tcga.

## Author Contributions

**Conceptualization:** Michael W. Greene, Elizabeth A. Lipke.

**Data curation:** Michael W. Greene, Peyton C. Kuhlers.

**Formal analysis:** Michael W. Greene, Peter T. Abraham, Peyton C. Kuhlers.

**Funding acquisition:** Michael W. Greene.

**Investigation:** Michael W. Greene, Peter T. Abraham, Peyton C. Kuhlers, Stanley T. Wijaya.

**Methodology:** Michael W. Greene, Peter T. Abraham, Peyton C. Kuhlers, Elizabeth A. Lipke.

**Project administration:** Michael W. Greene.

**Supervision:** Michael W. Greene, Elizabeth A. Lipke.

**Validation:** Michael W. Greene.

**Visualization:** Michael W. Greene, Martin J. Heslin.

**Writing – original draft:** Michael W. Greene, Peter T. Abraham, Stanley T. Wijaya, Ifeoluwa Odeniyi.

**Writing – review & editing:** Michael W. Greene, Peter T. Abraham, Peyton C. Kuhlers, Elizabeth A. Lipke, Martin J. Heslin, Stanley T. Wijaya, Ifeoluwa Odeniyi.

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
