## [Decision Letter · Decision Letter 0]

5 Jan 2022

PONE-D-21-24367Consensus molecular subtype differences linking colon adenocarcinoma and obesity revealed by a cohort transcriptomic analysisPLOS ONE

Dear Dr. Greene,

Thank you for submitting your manuscript to PLOS ONE. After careful consideration, we feel that it has merit but does not fully meet PLOS ONE’s publication criteria as it currently stands. Therefore, we invite you to submit a revised version of the manuscript that addresses the points raised during the review process. You will see that while the reviewers are persuaded of the importance of your study, they have raised several points that require consideration and revision. In particular, both reviews require clarification of some methodology and have raised several important points regarding discussion and interpretations of your results. In addition, reviewer 2 has suggested several further analyses to evaluate the robustness of your results, which should be addressed in your revised manuscript.

We look forward to receiving your revised manuscript.

Kind regards,

Katherine James, Ph.D.

Academic Editor

PLOS ONE

Journal Requirements:

Reviewers' comments:

Reviewer's Responses to Questions

**Comments to the Author**

1. Is the manuscript technically sound, and do the data support the conclusions?

Reviewer #1: Yes

Reviewer #2: Yes

2. Has the statistical analysis been performed appropriately and rigorously? 

Reviewer #1: Yes

Reviewer #2: Yes

3. Have the authors made all data underlying the findings in their manuscript fully available?

Reviewer #1: Yes

Reviewer #2: Yes

4. Is the manuscript presented in an intelligible fashion and written in standard English?

Reviewer #1: Yes

Reviewer #2: Yes

5. Review Comments to the Author

Reviewer #1: This paper explores associations of BMI status (obese/overweight/normal) with colorectal cancer (CRC) with respect to CMS and various gene expression summaries from TCGA. After identifying genes differentially expressed (DEG) based on BMI status for each CMS category, a GSEA is done to identify key subsets differing across BMI groups within each CMS, PPI networks and accompanying hub genes computed from these DEG, and comparing cancer vs. normal for these genes in other cohorts, a prognostic analysis comparing survival/regression free survival in groups upregulatred/not based on the PEG for BMI status, again separately for each CMS, and then drug sensitivity predictions are obtained based on BMI-selected PEG separately by CMS.

This paper provides a well-described detailed analysis of various aspects of CRC, characterizing differences based on BMI and separately within each CMS. This provides some potentially useful information for management of CRC.

However, the paper falls short in its underlying motivation. Among the unanswered questions of motivation are:

1. What are the authors hoping the reader to learn from obesity status in CRC? Various DEG are determined, and the corresponding pathways and hub genes, but how is this information to be validated and applied?

2. Why is it done separately by CMS? Is it thought obesi. ty has a different mechanism based on CMS?

The conclusion states that the paper showed "obesity differentially affected pathways, hub genes, survival and predicted drug sensitivity in CMS specific matter. But the authors never test whether these results are CMS specific -- they just assume they are and do CMS-separate analysis for each step -- none of which looks at statistical signifcance of the CMS modulatory effect.

3. Wha can we infer about the different hub genes by CMS, and what does the normal vs. cancer analysis of said genes show, and how is this useful?

4. The drug sensitivity analysis shows interesting hypothesis about potential obesity based precision therapeutics, yet this is not explicitly discussed, nor is it stated how results would be validated or translated. Are there cell line studies, e.g., that could be done to validate the obesity modulation appears to work?

Overall, this reads as a nice series of analyses, but it is not clear what the intended key resullts are and how these would be put into practice.

Other questions/comments:

* Some of these procedures (PROGgenev2) have tuning parameters, and several choices are arbitrary (20 genes for PROGgeneV2 and 10 genes for PPI). Please discuss how these tuning parameters were chosen and demonstrate sensitivity to their choice.

* The CMS3/obesity association is interesting. In light of the CMS3-race associations previously noted, it would be insightful to assess whether the black obese patients are more likely to be CMS3 than the other obese non-black patients.

* Some statements are given that are speculative and lack more precise statement -- e.g. "obesity may modulate the derivation of CMS3 and CMS4 tumors from canonical CMS2 tumors". This paper only looks at obesity, and doesn't consider other potential mechanisms or explanation -- so is just showing association. This may be a strong statement even with the "may modulate" qualifier.

Reviewer #2: The paper by Greene and colleagues reports interesting results of gene expression data by BMI categories, stratified by consensus-molecular subtypes (CMS) categories of colorectal cancer patients. A number of differentially expressed genes between obese/overweight and normal patients emerged, with some suggestions for differential associations with survival and drug sensitivity. The work is thorough and well conducted and explained. The manuscript is well written. My only concern is regarding the broad conclusions reached by investigating a relatively small sample set (especially within some CMS categories) and potential confounding, specifically by tumor stage, that needs to be addressed. Conclusions need to be less far-reaching.

Overall, a valuable contribution to the literature.

1) Abstract: please rephrase the conclusions to discuss “possible associations” rather than “effects”.

2) Patients: “Samples with an FDR greater than 0.05 were not classified”. Please add how many these were.

3) Considering the potential for confounding by race/ethnicity (e.g., Asian patients not represented among the obese), please perform a sensitivity analysis of the main findings, restricting the sample set to those from Caucasian patients.

4) Can results be adjusted for race/ethnicity?

5) Please add a table that illustrates what factors were associated with the CMS subtypes (similar to Table 1, but columns as CMS)

6) There could be confounding by tumor stage, especially because stage IV patients frequently present clinically after weight loss. Can the analysis of DEGs and GSEA be adjusted for tumor stage, to evaluate robustness? If not, can you exclude stage IV patients? Also, are CMS1 patients generally of lower age (and, accordingly, lower BMI) because they are more likely to have familial disease? If they are younger, please discuss this in the limitation section.

7) While intriguing, the results of DEGs may also be somewhat random. Please be more cautious in the interpretation.

8) Consider streamlining the text description of the GSEA results

9) It appears that most of the signals are appearing in CMS4 and less CMS3. This points toward greater impact of inflammation in the adipose tissue as a driver, rather than metabolic differences. The authors might want to consider making this point more clearly

10) Discussion: “These findings suggest a differential role of inflammation in the subtypes… whether the patient is overweight or obese” – from my read of the data it looks more like there was no substantial distinction between overweight and obese (e.g., in the direct comparison), but clearly a difference to normal. Perhaps other data are meant? Please clarify.

11) Paragraph “The deregulation of cellular energetics” should be worded more cautiously, because this was limited only to the CMS4 subtype.

12) Sentence “obesity may modulate the derivation of CMS3 and CMS4 tumors from canonical CMS2 tumors”. The evidence appears to be much stronger for CMS4. Please make that distinction clear. Overall, CMS3 does not emerge with strong signals from what I see?

13) Conclusion: Please rephrase first sentence to avoid causality (e.g., to “Our findings suggest that obesity is associated with CMS-specific CRC tumors”) and overall reduce the claims made in the conclusions, considering the limitations of the study.

14) Figure 3: Please make clear (including in the legend) why not all CMS are shown. These results are based on small numbers and should be interpreted with caution.

Minor comments:

1) Some additional references to add

a. to refs 22-24 on adipose tissue and other mechanisms of energy balance and gastrointestinal cancer: Ulrich et al. Nat Rev Gastroenterol Hepatol. 2018;15:683-98.

b. Also: Haffa et al: J Clin Endocrinol Metab. 2019;1;104:5225-37

2) DEG analysis: there is a repetition of “DEGs DEGs”

3) CMS hub genes: Word missing “To examine the CRC relevance… compiled from THE NCBI”

4) The quality of the figures needs to be improved.

5) Figure 1 should read Euler, not Eular

6. PLOS authors have the option to publish the peer review history of their article (what does this mean?). If published, this will include your full peer review and any attached files.

Reviewer #1: No

Reviewer #2: No

---

## [Author Response · Author response to Decision Letter 0]

4 Mar 2022

Responses to reviewers

We thank the reviewers for their helpful comments that have led to an improved manuscript. Please see the Response to reviewers document for a color version of the response where our response to each comment is written in a blue font. Changes to the manuscript text are written in red font.

Reviewer #1: This paper explores associations of BMI status (obese/overweight/normal) with colorectal cancer (CRC) with respect to CMS and various gene expression summaries from TCGA. After identifying genes differentially expressed (DEG) based on BMI status for each CMS category, a GSEA is done to identify key subsets differing across BMI groups within each CMS, PPI networks and accompanying hub genes computed from these DEG, and comparing cancer vs. normal for these genes in other cohorts, a prognostic analysis comparing survival/regression free survival in groups upregulatred/not based on the PEG for BMI status, again separately for each CMS, and then drug sensitivity predictions are obtained based on BMI-selected PEG separately by CMS.

This paper provides a well-described detailed analysis of various aspects of CRC, characterizing differences based on BMI and separately within each CMS. This provides some potentially useful information for management of CRC.

However, the paper falls short in its underlying motivation. Among the unanswered questions of motivation are:

1. What are the authors hoping the reader to learn from obesity status in CRC? Various DEG are determined, and the corresponding pathways and hub genes, but how is this information to be validated and applied?

We have revised the manuscript to more clearly state what will be gained from our findings. The following sentence written in red text was added at the end of the introduction: 

Thus, there is compelling epidemiological and experimental evidence linking obesity to CRC (9-12) – although more strongly for colon cancer (9, 11). The obesity-cancer link is thought to be driven by multiple obesity-derived factors that activate pathways mediating cell signaling, proliferation, and tumor progression (38). Yet, there does not exist a framework for activation of these obesity-driven cell pathways. Thus, we questioned whether the effect of obesity on cell signaling, proliferation, and tumor progression pathways in CRC tumors is dependent on the CMS of the tumor. Therefore, we undertook a study to examine the transcriptomic profile of colon adenocarcinoma tumors from obese patients compared to healthy and overweight BMI patients to determine whether obesity modulates cell signaling, proliferation, and tumor progression pathways in a similar manner across the four CMSs. We also examined whether prognostic survival outcomes and predicted drug response in obesity associated differentially expressed genes (DEGs) is similar between the four CMSs. Our secondary objective was to determine whether the transcriptomic profile of tumors from overweight patients compared to healthy BMI patients is similar between the four CMSs. The knowledge gained from our findings can be used to test using in vitro and in animal models whether key genes and pathways link obesity in a CMS-dependent manner to identify new therapeutic targets to treat colon cancer.

We have added following text written in red to the Conclusions section: 

We conclude that obesity has CMS-specific effects in colon cancer. This conclusion is based not only on CMS-specific effects of obesity in gene set enrichment but also on findings of obesity-related DEGs and the identification of unique hub genes for each CMS in tumors from obese patients. Prognostic patient survival analysis and predicted drug sensitivity support our findings that obesity has CMS-specific effects in colon cancer. Taken together, our findings are consistent with the hypothesis that the obesity-cancer link is mediated by obesity-derived factors which converge on key cell signaling and metabolic pathways; yet this occurs in a CMS-specific manner in colon cancer. These findings will require validation using in vitro and animal models to examine the CMS-dependence of the genes and pathways. Once validated the obesity-linked genes and pathways may represent new therapeutic targets to treat colon cancer in a CMS-dependent manner.

We have added following text written in red to the Abstract:

Our findings support that obesity impacts the CRC tumor transcriptome in a CMS-specific manner. The possible associations reported here will require validation using in vitro and animal models to examine the CMS-dependence of the genes and pathways. Once validated the obesity-linked genes and pathways may represent new therapeutic targets to treat colon cancer in a CMS-dependent manner.

2. Why is it done separately by CMS? Is it thought obesi. ty has a different mechanism based on CMS?

The conclusion states that the paper showed "obesity differentially affected pathways, hub genes, survival and predicted drug sensitivity in CMS specific matter. But the authors never test whether these results are CMS specific -- they just assume they are and do CMS-separate analysis for each step -- none of which looks at statistical significance of the CMS modulatory effect.

There are currently no reports that obesity modulates the tumor transcriptome based on CMS. We specifically questioned whether obesity modulates cell signaling, proliferation, and tumor progression pathways in a similar manner across the four CMSs. Thus, we undertook an analysis to assess the effect of obesity within each CMS. 

To further examine the impact of obesity we have performed two new analyses to directly test the effect of obesity across the CMSs. In the first analysis we used an interaction term for obesity:CMS in the DESeq2 linear model followed by pairwise comparisons between CMSs. Consistent with our findings of BMI comparisons within each CMS, we observed that there was a unique pattern of overlapping DEGs for each CMS. This new data is presented in Supplementary Figure 2. The follow text has been added to the Results section:

To directly examine the impact of obesity across CMSs, we used an interaction term for obesity:CMS in the DESeq2 linear model followed by pairwise comparisons between CMSs. The Euler diagrams of DEGs demonstrate a unique pattern of overlapping DEGs for each CMS (Suppl. Fig 2). The greatest overlap in DEGs across CMS comparisons was observed from CMS1 (18 gene) while the least overlap was observed in CMS4 (1 gene) (Suppl. Fig 2).

The second new analysis we performed was to use the Likelihood ratio test in DESeq2 to identify the obesity effect across CMSs. We observed 1579 obesity-linked DEGs (padjusted < 0.05; BaseMean > 10; >1 log2FoldChange). We constructed a Protein-Protein Interaction (PPI) network using the STRING database module in Cytoscape. From the PPI network, hub genes were identified using the MCC algorithm of the CytoHubba module in Cytoscape. An inflammation-linked gene hub was identified: FCGR3A, IL10, CXCL8, ITGAM, CD86, CXCL10, PTPRC, CCR2, CCR5, CCL4. This inflammation-linked gene hub represents a general effect of obesity on the CRC transcriptome and supports our findings that obesity has CMS-specific transcriptomic effects. This new data is presented in Supplementary Figure 6 and discussed in the results and discussion sections. 

Results section: 

Obesity specific hub genes across CMS categories 

To directly examine the impact of obesity across CMSs, we used the Likelihood ratio test in DESeq2, which tests a reduced model with all assigned and unassigned CMS tumor samples, to identify obesity-linked DEGs. We observed 1579 obesity-linked DEGs (p adjusted < 0.05; BaseMean > 10; >1 log2FoldChange). We constructed a PPI network from the DEGs using the STRING database and identified hub genes using the MCC algorithm and confirmed in at least 3 other topographical algorithms in CytoHubba as described above. An inflammation gene hub was identified. Hub genes commonly identified were Fc region receptor III-A (FCGR3A), IL10, C-X-C Motif Chemokine Ligand 8 (CXCL8), Integrin Subunit Alpha M (ITGAM), Cluster of Differentiation 86 (CD86), Protein Tyrosine Phosphatase Receptor Type C (PTPRC), and C-C Motif Chemokine Receptor 5 (CCR5) (Suppl. Fig. 7). This gene set represents a general effect of obesity on the CRC transcriptome that is not specific for any one CMS.

Discussion section:

The commonality of immune-related Hallmark gene set enrichment in 3 of the 4 subtypes and our finding of an inflammation-linked gene hub across CMSs are important because an inflammatory risk score is an independent predictor for stage II colon cancer prognosis (68) and a circulating inflammation signature is a strong prognostic factor of progression-free and overall survival of patients with metastatic CRC (29). Consistent with these findings, a higher dietary inflammatory potential is associated with higher CRC risk (69).

The follow text in red has been added to the Methods section:

Transcriptomic Analysis:

The R package DESeq2 (44) was used to assess differential gene expression. Raw counts and associated phenotypes were inputted into DESeq2, and the following contrasts were made within each CMS category: obese vs. normal, obese vs. overweight, and overweight vs. normal. Additionally, the effect of obesity between the subtypes was evaluated by adding an interaction term to the design, which allowed for comparison between individual CMSs using specified contrasts and for comparison across the CMSs using a likelihood ratio test. Genes with a base mean expression greater than ten and an adjusted p-value less than 0.05 were used for downstream analysis and visualization Volcano plots were generated using the R package EnhancedVolcano [https://github.com/kevinblighe/EnhancedVolcano]. Gene overlap between comparisons were visualized using Euler diagrams and upset plots from the R packages eulerr [https://github.com/jolars/eulerr] and UpsetR [6], respectively.

3. Wha can we infer about the different hub genes by CMS, and what does the normal vs. cancer analysis of said genes show, and how is this useful?

We can infer from our CMS-specific hub gene findings that there may exist not yet considered mechanisms playing a role in modulating the colon cancer tumor transcriptome. The CMS-specific hub gene findings are useful for hypothesis testing to examine new colon cancer mechanisms and identify new therapeutic targets to treat colon cancer. 

We have added following text to the Results section: 

… Finally, CCR2 gene expression was significantly (p < 0.001) downregulated (Log2 fold change = -0.706) in colon cancer (Fig 6E). Taken together our analysis of CMS-specific hub genes infers that there may exist new mechanisms playing a role in modulating the colon cancer tumor transcriptome. The CMS-specific hub gene findings are useful for hypothesis testing to examine new colon cancer mechanisms and identify new therapeutic targets to treat colon cancer. 

The normal vs. cancer analysis was undertaken to examine whether the genes are relevant to colon cancer. A limitation of these findings is that they do not assess whether the expression of the genes is obesity-linked. However, given the high prevalence of obesity in many populations throughout the world, our findings suggest that obesity may be affecting the colon cancer tumor transcriptome. 

4. The drug sensitivity analysis shows interesting hypothesis about potential obesity based precision therapeutics, yet this is not explicitly discussed, nor is it stated how results would be validated or translated. Are there cell line studies, e.g., that could be done to validate the obesity modulation appears to work?

We thank the reviewer for suggesting additional text to discuss the drug sensitivity analysis findings. We have added the following text written in red to the Results section: 

… significantly increased predicted drug sensitivity for the drugs targeting EGFR and significantly reduced predicted drug sensitivity for a drug targeting WNT signaling were observed (Suppl. Table 3). Taken together, these findings suggest that there is a potential for obesity-based precision therapeutics. However, our findings will require validation with both in vitro drug testing using cell models of obesity-linked inflammation and insulin resistance and drug studies in obese animal models using CMS-specific patient-derived xenografts prior to clinical evaluation.

Overall, this reads as a nice series of analyses, but it is not clear what the intended key resullts are and how these would be put into practice.

We have clarified our overall conclusion with the following sentences in the Conclusions:

… Taken together, our findings are consistent with the hypothesis that the obesity-cancer link is mediated by obesity-derived factors which converge on key cell signaling and metabolic pathways; yet this occurs in a CMS-specific manner in colon cancer. These findings will require validation using in vitro and animal models to examine the CMS-dependence of the genes and pathways. Once validated the obesity-linked genes and pathways may represent new therapeutic targets to treat colon cancer in a CMS-dependent manner.

Other questions/comments:

* Some of these procedures (PROGgenev2) have tuning parameters, and several choices are arbitrary (20 genes for PROGgeneV2 and 10 genes for PPI). Please discuss how these tuning parameters were chosen and demonstrate sensitivity to their choice.

The following text (in red) was added to the Methods section to provide justification for using 20 genes in the PROGgenev2 analysis: 

was assessed using the PROGgeneV2 tool was used (56, 57). We chose to use 20 genes because an approximately 20 gene set can distinguish CRC patients with low or high risk of disease relapse (58) and a 20 gene set has prognostic value for overall survival in CRC patients when adjusted for age, gender, and stage (59).

The following citations were added to the References section: 

Kopetz S, Tabernero J, Rosenberg R, Jiang ZQ, Moreno V, Bachleitner-Hofmann T, et al. Genomic classifier ColoPrint predicts recurrence in stage II colorectal cancer patients more accurately than clinical factors. Oncologist. 2015;20(2):127-33

Barriuso J, Nagaraju RT, Belgamwar S, Chakrabarty B, Burghel GJ, Schlecht H, et al. Early Adaptation of Colorectal Cancer Cells to the Peritoneal Cavity Is Associated with Activation of “Stemness” Programs and Local Inflammation. Clinical Cancer Research. 2021;27(4):1119-30

Unfortunately, the PROGgeneV2 tool is no longer available to the public. Thus, we were unable to test survival analysis using different size gene sets.

We selected 10 hub genes using the Maximal Clique Centrality (MCC) algorithm because the MCC was found to have better performance on the precision of predicting essential proteins in a model PPI network compared to other topological algorithms (1).

1. Chin C-H, Chen S-H, Wu H-H, Ho C-W, Ko M-T, Lin C-Y. cytoHubba: identifying hub objects and sub-networks from complex interactome. BMC Systems Biology. 2014 2014/12/08;8(4):S11.

To demonstrate sensitivity of our choice for 10 hub genes, we compared the hub genes identified using the Maximal Clique Centrality (MCC) topological algorithm to the four other topological algorithms: Degree, Edge Percolated Component (EPC), Maximum Neighborhood Component (MNC), and Density of Maximum Neighborhood Component (DMNC). In this new analysis, hub genes commonly identified in at least 4 out of the 5 topographical algorithms are shown in Supplemental Table 3. We revised Figure 5 to indicate common hub genes across the algorithms. 

We have added following text written in red to the Results section:

To gain insight into obesity-regulated hub genes within each CMS group, we first constructed a Protein-Protein Interaction (PPI) network using the STRING database module in Cytoscape. From the PPI network, hub genes were identified using the MCC algorithm of the CytoHubba module in Cytoscape. The top ten highest scoring genes in the obese to normal BMI patients and the obese to overweight comparisons for each CMS are shown in Figure 5 and Supplemental Table 2. A sensitivity analysis of the MCC algorithm hub genes was performed using four other topographical algorithms. Hub genes commonly identified in at least 4 out of the 5 topographical algorithms are shown in Supplemental Table 3. We observed that in CMS1 obese to normal BMI comparison there were four hub genes: Bassoon presynaptic cytomatrix protein (BSN), Major synaptic vesicle protein p38 (SYP), and RAB3C, member RAS oncogene family (RAB3C), and Unc-13 homolog A (UNC13A). In contrast, an immune hub containing interleukin 10 (IL-10), C-C motif chemokine receptor 2 (CCR2), and C-C motif chemokine ligand 13 (CCL13) was observed in the obese to overweight BMI comparison for CMS1. Weak interconnectivity was observed in the obese to normal BMI comparison for CMS2, while a hub containing NK2 homeobox 1 (NKX2-1) and SRY-box transcription factor 2 (SOX2) was observed in the obese to overweight BMI comparison for CMS2. A hub containing four Melanoma-associated antigen genes (MAGEA6, MAGEA3, MAGEA11, and MAGEA12) was observed in the obese to normal BMI comparison for CMS3, while an interconnected hub network of genes including a somatostatin receptor gene (SSTR5) but also a C-C motif chemokine receptor genes (CCR2) was observed in the obese to overweight BMI comparison for CMS3. In the obese to normal comparison for CMS4 we observed a hub with UDP glucuronosyltransferase 1 family, polypeptide genes (UGT1A1 and UGT1A8), while a hub network of genes including neuromedin U receptor 2 (NMUR2), peptide YY (PYY) and pro-platelet basic protein (PPBP) was observed in the obese to overweight BMI comparison for CMS4. The MCC algorithm hub genes for the overweight to normal BMI comparison were also identified (Suppl. Fig 5 and Suppl. Table 3). Hub genes commonly identified in at least 4 out of the 5 topographical algorithms are shown in Supplemental Table 3. In CMS1 there was an overlap in SYP with the hub observed in the obese comparison to normal BMI while no overlap in hub genes was observed for CMS2. An overlap in the Melanoma-associated antigen genes (MAGEA6, MAGEA3, and MAGEA12) was observed with the hub observed in the CMS3 obese comparison to normal BMI, while no overlaps were observed in CMS4. 

* The CMS3/obesity association is interesting. In light of the CMS3-race associations previously noted, it would be insightful to assess whether the black obese patients are more likely to be CMS3 than the other obese non-black patients.

As suggested by the reviewer were performed a chi-square analysis with the obese patients to examine the interaction between race and CMS categories. We found that the percentage of Black or African American patients across the CMS categories was significantly different.

 BLACK OR AFRICAN AMERICAN WHITE

 CMS1 0 100

 CMS2 36 64

 CMS3 50 50

 CMS4 17 80

The following new findings have been added to the Results section: 

… In contrast, the CMS classification of tumors was significantly different across the BMI categories (p = 0.040): a greater proportion of CMS3 tumors was observed in the obese (22%) compared to the normal (4%) BMI category. Consistent with this observation, the highest average BMI was in the CMS3 group (31.2) followed CMS4 (28.6), CMS1 (27.4), and CMS2 (27.1). We also observed in obese patients a significant difference (p = 0.023) in the percentage of Black or African American patients across the CMS categories: CMS3 had the highest percentage (50%) and CMS1 (0%) had the lowest of obese Black or African American patients.

* Some statements are given that are speculative and lack more precise statement -- e.g. "obesity may modulate the derivation of CMS3 and CMS4 tumors from canonical CMS2 tumors". This paper only looks at obesity, and doesn't consider other potential mechanisms or explanation -- so is just showing association. This may be a strong statement even with the "may modulate" qualifier.

We have revised the Discussion section to delete the statement quoted above and to indicate that it is not known whether obesity plays a role in shifting CMS2 tumors to CMS3 or CMS4. 

We have added following text written in red to the Discussion section:

It has been proposed that a metabolic shift in the canonical CMS2 tumors possibly due to KRAS mutations and copy number events results in CMS3 tumors whereas the stromal-enriched inflamed tumor microenvironment is the driver for the development of CMS4 tumors from the CMS2 subtype (5). Interestingly, we observed an obesity-induced enrichment of EMT- and metabolism-related Hallmark gene sets in CMS4 tumors, and that a greater proportion of CMS3 tumors in obese compared to normal BMI patients. Our later finding is consistent with a report (76) that patients with CMS3 tumors in a Stage II-IV CRC cohort are more likely (OR 3.5, 95% CI 1.1-11.4) to have type 2 diabetes, an obesity-linked disease. Whether obesity plays a role in shifting CMS2 tumors to CMS3 or CMS4 is not known.

Reviewer #2: The paper by Greene and colleagues reports interesting results of gene expression data by BMI categories, stratified by consensus-molecular subtypes (CMS) categories of colorectal cancer patients. A number of differentially expressed genes between obese/overweight and normal patients emerged, with some suggestions for differential associations with survival and drug sensitivity. The work is thorough and well conducted and explained. The manuscript is well written. My only concern is regarding the broad conclusions reached by investigating a relatively small sample set (especially within some CMS categories) and potential confounding, specifically by tumor stage, that needs to be addressed. Conclusions need to be less far-reaching.

Overall, a valuable contribution to the literature.

1) Abstract: please rephrase the conclusions to discuss “possible associations” rather than “effects”.

The Abstract has been revised to make the conclusion less far-reaching:

Conclusions

Our findings support that obesity impacts the CRC tumor transcriptome in a CMS-specific manner. The possible associations reported here will require validation using in vitro and animal models to examine the CMS-dependence of the genes and pathways.

2) Patients: “Samples with an FDR greater than 0.05 were not classified”. Please add how many these were.

We have added following text written in red to the Methods section:

Twenty-three patient samples (10%) with an FDR greater than 0.05 were not assigned to a CMS.

3) Considering the potential for confounding by race/ethnicity (e.g., Asian patients not represented among the obese), please perform a sensitivity analysis of the main findings, restricting the sample set to those from Caucasian patients.

We have performed this sensitivity analysis using the Obese vs Normal GSEA findings. The new findings are reported in Supplemental Figure 4. We found that the significantly enriched Hallmark gene sets in the original analysis when compared to the data analysis excluding the Asian patients was highly correlated. The result from the linear regression analysis is:

Coefficients:

 Estimate Std. Error t value Pr(>|t|) 

OBvsN_NES 1.001165 0.108337 9.241 4.97e-09 ***

We have added following text written in red to the Results section:

A sensitivity analysis was performed to determine whether inclusion of Asian patients, which were all categorized as normal BMI, affected gene set enrichment analysis comparing RNA seq data from obese to normal BMI patients for each CMS. As shown in Supplemental Figure 4, the enrichment of Hallmark gene sets in obese patients mirrored the findings observed with whole patient population (Fig. 4). All 27 enriched gene sets in the whole population were enriched in the sensitivity analysis. Further, the normalized enrichment score was highly correlated (β = 1.001, SE = 0.108, p = 5.0 x 10-9) between the whole patient population and the population excluding Asians. However, the exclusion of Asians in the sensitivity analysis did result in significant enrichment of three EMT gene sets in obese CMS4 tumors that did not reach significance in the whole patient population.

4) Can results be adjusted for race/ethnicity?

Based on our finding from the sensitivity analysis, we believe that race/ethnicity is not altering our findings.

5) Please add a table that illustrates what factors were associated with the CMS subtypes (similar to Table 1, but columns as CMS)

The new table has been added as Supplemental Table 1. The following text has been added to the Results section to report the findings in the table:

The patient’s demographic and clinical tumor data was also assessed across the CMS categories (Suppl. Table 1). No significant differences in sex, age, ethnicity, race, tumor stage, or lymph node ratio were observed across the CMS categories. In contrast, tumor location was significantly different across the CMS categories (p = 0.004). Consistent with the assessment of the patient’s demographic and clinical tumor data assessed across the BMI categories, the BMI classification of patients was significantly different across the CMS categories (p = 0.040).

6) There could be confounding by tumor stage, especially because stage IV patients frequently present clinically after weight loss. Can the analysis of DEGs and GSEA be adjusted for tumor stage, to evaluate robustness? If not, can you exclude stage IV patients? Also, are CMS1 patients generally of lower age (and, accordingly, lower BMI) because they are more likely to have familial disease? If they are younger, please discuss this in the limitation section.

Even though we did not observe significant differences in Tumor Stage across the BMI categories, it is still possible that there could be confounding by weight loss in Stage IV patients. We have included this limitation in the discussion. For patients under 50 years old, we do not observe any pattern across the CMS categories (new data presented in Supplementary Table 1). 

We have added following text written in red to the Discussion section:

A limitation of the current study is that it was only performed in TCGA-COAD cohort which lacked racial diversity. CMS categorization of CRC patients across the BMI categories led to small samples sizes which lowered statistical power for some of the comparisons particularly those with normal BMI patients with CMS3 tumors. Confirmation of our findings will require assembly of a large CRC cohort that contains both transcriptomic and body weight and height data. Another limitations of the current study are that confounding factors that are risk factors for CRC such as smoking and alcohol consumption were not assessed and disease associated weight loss across the CMS categories was not assessed. Finally, it should be noted that BMI is a proxy for adiposity but may not account for the metabolic health of the patients. 

7) While intriguing, the results of DEGs may also be somewhat random. Please be more cautious in the interpretation.

We agree with the reviewer and have focused our attention only on genes in which overlaps were observed across CMS categories. In addition, to demonstrate sensitivity of our choice for 10 hub genes, we compared the hub genes identified using the Maximal Clique Centrality (MCC) topological algorithm to the four other topological algorithms: Degree, Edge Percolated Component (EPC), Maximum Neighborhood Component (MNC), and Density of Maximum Neighborhood Component (DMNC). In this new analysis, hub genes commonly identified in at least 4 out of the 5 topographical algorithms are shown in Supplemental Table 3. We revised Figure 5 to indicate common hub genes across the algorithms.

8) Consider streamlining the text description of the GSEA results

We have reduced the GSEA results section by 81 words. However, we did add the sensitivity analysis to this section as suggested above.

9) It appears that most of the signals are appearing in CMS4 and less CMS3. This points toward greater impact of inflammation in the adipose tissue as a driver, rather than metabolic differences. The authors might want to consider making this point more clearly

The reviewer makes an interesting point. We agree that our data points to prominent obesity associated changes in the CMS4 tumor transcriptome. The importance of adipose inflammation versus metabolic differences is an intriguing hypothesis. However, our analysis of the CMS3 tumor transcriptome was limited to only 3 patients with a normal BMI. Because of this limitation, we are uncomfortable speculating on the importance of adipose inflammation versus metabolic differences.

10) Discussion: “These findings suggest a differential role of inflammation in the subtypes… whether the patient is overweight or obese” – from my read of the data it looks more like there was no substantial distinction between overweight and obese (e.g., in the direct comparison), but clearly a difference to normal. Perhaps other data are meant? Please clarify.

The differences observed between overweight and obese patients was dependent on the CMS. For example, Inflammation-related Hallmark gene sets were in enriched in both the Obese vs Normal and Obese vs Overweight CMS1 comparisons, but that was not observed in the CMS4 comparisons. Further, we observed differences in each of our assessments for the Obese vs Overweight comparisons. Thus, our data suggests that BMI associations to the tumor transcriptome is specific for each CMS and that there are differences between Obese and Overweight patients.

11) Paragraph “The deregulation of cellular energetics” should be worded more cautiously, because this was limited only to the CMS4 subtype.

The enrichment of Metabolism-related Hallmark gene sets was indeed concentrated in CMS4 tumors. However, we did observe Myc target gene sets enriched in CMS2 and CMS3 tumors. One limitation of our categorization of Hallmark gene sets is that the gene sets can overlap (e.g. Myc targets are reported in the Cell Cycle-related gene sets but Myc targets also regulate metabolism, particularly glucose metabolism). We have revised the paragraph to more accurately discuss the data. We have revised the following text written in red in the Discussion section:

The deregulation of cellular energetics resulting in the reprogramming of energy metabolism plays a role in tumorigenesis (72) and has been identified as an emerging hallmark of cancer (73). Our observation that CMS3 and CMS4 enrichment in the Myc gene sets Myc targets V1 and Myc targets V2 suggests that key cell signaling and metabolic pathways within tumor cells driving tumor growth and progression are differentially regulated by obesity. It has been hypothesized that obesity-derived factors (e.g. circulating hormones, adipokines, inflammatory cytokines, and dietary factors) converge on these key cell signaling and metabolic pathways (38). The enrichment of metabolism related gene sets was concentrated in CMS4 tumors. Consistent with this finding, we also observed that an obesity-linked network of hub genes related to the UGT1A gene locus was downregulated in CMS4 tumors. In agreement with this observation, we found that UGT1A1, UGT1A6, and UGT1A8 expression was reduced in colon adenocarcinoma compared to normal colon tissue in 3 separate large CRC cohorts. The UGT1 subfamily of enzymes reduces the biological activity and enhances the solubility of lipophilic substrates through the process of glucuronidation (74). UGT activity has been hypothesized to modulate energy metabolism by altering cellular pools UDP-sugars which are glycolytic intermediates or through interaction with pyruvate kinase (PKM2), a glycolytic enzyme (74, 75).

12) Sentence “obesity may modulate the derivation of CMS3 and CMS4 tumors from canonical CMS2 tumors”. The evidence appears to be much stronger for CMS4. Please make that distinction clear. Overall, CMS3 does not emerge with strong signals from what I see?

We have revised the Discussion section to delete the statement quoted above and to indicate that it is not known whether obesity plays a role in shifting CMS2 tumors to CMS3 or CMS4 is not known. 

We have added following text written in red to the Discussion section:

It has been proposed that a metabolic shift in the canonical CMS2 tumors possibly due to KRAS mutations and copy number events results in CMS3 tumors whereas the stromal-enriched inflamed tumor microenvironment is the driver for the development of CMS4 tumors from the CMS2 subtype (5). Interestingly, we observed an obesity-induced enrichment of EMT- and metabolism-related Hallmark gene sets in CMS4 tumors, and that a greater proportion of CMS3 tumors in obese compared to normal BMI patients. Our later finding is consistent with a report (76) that patients with CMS3 tumors in a Stage II-IV CRC cohort are more likely (OR 3.5, 95% CI 1.1-11.4) to have type 2 diabetes, an obesity-linked disease. Whether obesity plays a role in shifting CMS2 tumors to CMS3 or CMS4 is not known.

13) Conclusion: Please rephrase first sentence to avoid causality (e.g., to “Our findings suggest that obesity is associated with CMS-specific CRC tumors”) and overall reduce the claims made in the conclusions, considering the limitations of the study.

We have revised the following text written in red in the Discussion section:

Our findings suggest that obesity may have CMS-specific associations in colon cancer.

We have revised the following text written in red in the Abstract:

Our findings suggest that obesity may have CMS-specific associations on the CRC tumor transcriptome.

14) Figure 3: Please make clear (including in the legend) why not all CMS are shown. These results are based on small numbers and should be interpreted with caution.

We have revised the following text written in red in the Figure 3 legend:

Figure 3. Prognostic patient outcomes reveal CMS-specific differences between obese BMI groups. The expression of the top 20 significantly upregulated DESeq2-obtained DEGs (MeanBase > 10, FDR p value < 0.05) in the obese vs. normal and obese vs. overweight comparisons for each CMS category were assessed in GSE17536 and GSE41258 for overall survival (A) and GSE14333 and GSE17536 for relapse-free survival (B) using the PROGgeneV2 tool. Survival analyses were adjusted for age, stage, and gender covariates and bifurcated based on median expression. The hazard ratios, 95% confidence intervals, and p values were reported for the Kaplan-Meier plots. Only statistically significant findings are shown and should be interpreted with caution. 

Minor comments:

1) Some additional references to add

a. to refs 22-24 on adipose tissue and other mechanisms of energy balance and gastrointestinal cancer: Ulrich et al. Nat Rev Gastroenterol Hepatol. 2018;15:683-98.

b. Also: Haffa et al: J Clin Endocrinol Metab. 2019;1;104:5225-37

These references have been added.

2) DEG analysis: there is a repetition of “DEGs DEGs”

3) CMS hub genes: Word missing “To examine the CRC relevance… compiled from THE NCBI”

4) The quality of the figures needs to be improved.

5) Figure 1 should read Euler, not Eular

We thank the reviewer for catching the typos listed above. These have been corrected. The figures were assembled in Adobe Illustrator and now saved as .eps files. For some reason the eps files for fig 2 and fig 4 are rotated in the pdf generated by PLOS One. If the eps files are downloaded, they are in the correct orientation.

---

## [Decision Letter · Decision Letter 1]

4 Apr 2022

PONE-D-21-24367R1Consensus molecular subtype differences linking colon adenocarcinoma and obesity revealed by a cohort transcriptomic analysisPLOS ONE

Dear Dr. Greene,

Thank you for submitting your manuscript to PLOS ONE. After careful consideration, we feel that it has merit but does not fully meet PLOS ONE’s publication criteria as it currently stands. Therefore, we invite you to submit a revised version of the manuscript that addresses the points raised during the review process.

 You will see that while reviewer 1 is happy with your updated manuscript, reviewer 2 has several outstanding concerns, which we need you to address. In particular, the results interpretation should discuss the limitations of the study, especially in relation to confounders, and additional analyses have been requested to aid the understanding of confounder effects.

We look forward to receiving your revised manuscript.

Kind regards,

Katherine James, Ph.D.

Academic Editor

PLOS ONE

Reviewers' comments:

Reviewer's Responses to Questions

**Comments to the Author**

1. If the authors have adequately addressed your comments raised in a previous round of review and you feel that this manuscript is now acceptable for publication, you may indicate that here to bypass the “Comments to the Author” section, enter your conflict of interest statement in the “Confidential to Editor” section, and submit your "Accept" recommendation.

Reviewer #1: All comments have been addressed

Reviewer #2: (No Response)

2. Is the manuscript technically sound, and do the data support the conclusions?

Reviewer #1: Yes

Reviewer #2: Partly

3. Has the statistical analysis been performed appropriately and rigorously? 

Reviewer #1: Yes

Reviewer #2: Yes

4. Have the authors made all data underlying the findings in their manuscript fully available?

Reviewer #1: Yes

Reviewer #2: Yes

5. Is the manuscript presented in an intelligible fashion and written in standard English?

Reviewer #1: Yes

Reviewer #2: Yes

6. Review Comments to the Author

Reviewer #1: Thank you for your responsive revision that has produced a greatly improved manuscript, making a nice contribution to the CRC and CMS literature..

Reviewer #2: Review of Greene et al

The authors have completed a series of additional analyses to strengthen the manuscript and justify their conclusions. There are unfortunately still some unclear aspects that need to be evaluated and discussed.

1) While reviewer 3 commented on the need to be more cautious in interpretation and refer to “associations” rather than effects in this cross-sectional analysis, there are still numerous sentences and the main conclusions that discuss “effects”. E.g., the conclusions start way too strong: “We conclude that obesity has CMS-specific effects in colon cancer”.

2) This is a first, preliminary analysis and not everything can be perfect. But please interpret the findings as such, with more caution, throughout the manuscript.

3) Reviewer 3 comment 3. Please provide the results with and without Asians side by side to enable the comparison. It seems like there is not corresponding figure for the overweight to normal among the full sample set.

4) Comment 3 is critical to understand the impact of race/ethnicity on study results, due to substantial possible confounding. Please acknowledge this limitation also in the discussion.

5) Comment 6 from reviewer 3 was either not understood or not done. Please add analyses removing stage IV, if adjustment is not possible.

6) Reviewer 1 comment 3: The response starting with “The normal vs cancer analysis…” is meaningless and not relevant to the paper.

7) Please understand that CMS1 is most likely familial disease and patients are significantly younger. This needs to be added as a limitation, along with the racial composition.

Despite these shortcomings of the study I remain enthusiastic of this publication. Please simply be mindful of the many limitations in the work.

7. PLOS authors have the option to publish the peer review history of their article (what does this mean?). If published, this will include your full peer review and any attached files.

Reviewer #1: **Yes: **Jeffrey S Morris

Reviewer #2: No

---

## [Author Response · Author response to Decision Letter 1]

13 Apr 2022

Note: The Response to reviewers document in the manuscript has the blue and red text.

Review of Greene et al

The authors have completed a series of additional analyses to strengthen the manuscript and justify their conclusions. There are unfortunately still some unclear aspects that need to be evaluated and discussed.

We thank the reviewer for their helpful comments that have led to an improved manuscript. Our response to each comment is written in a blue font. Changes to the manuscript text are written in red font below.

1) While reviewer 3 commented on the need to be more cautious in interpretation and refer to “associations” rather than effects in this cross-sectional analysis, there are still numerous sentences and the main conclusions that discuss “effects”. E.g., the conclusions start way too strong: “We conclude that obesity has CMS-specific effects in colon cancer”.

We have gone through the manuscript and have identified three instances where “effects” were stated. All three have been revised to “associations”: once in the Results section of the Abstract, and twice in the Conclusions.

2) This is a first, preliminary analysis and not everything can be perfect. But please interpret the findings as such, with more caution, throughout the manuscript.

To emphasis the preliminary nature of the findings we have added text to the conclusions in the Abstract: Our findings support that obesity impacts the CRC tumor transcriptome in a CMS-specific manner. The possible associations reported here are preliminary and will require validation using in vitro and animal models to examine the CMS-dependence of the genes and pathways. Once validated the obesity-linked genes and pathways may represent new therapeutic targets to treat colon cancer in a CMS-dependent manner.

We have also added “preliminary findings” to seven paragraphs in the results section where findings have been summarized.

3) Reviewer 3 comment 3. Please provide the results with and without Asians side by side to enable the comparison. It seems like there is not corresponding figure for the overweight to normal among the full sample set.

We have performed the overweight to normal analysis for the cohort minus Asians. The new data is now shown side-by-side with the GSEA findings from the full cohort (Supplementary Figure 5). To describe the new data we have added the following text to the Results section: A sensitivity analysis was performed to determine whether inclusion of Asian patients, which were all categorized as normal BMI, affected gene set enrichment analysis comparing RNA seq data from obese to normal and overweight to normal BMI patients for each CMS. As shown in Supplemental Figure 4B, the enrichment of Hallmark gene sets in obese patients mirrored the findings observed with whole patient population (Fig. 4A). All 27 enriched gene sets in the whole population were enriched in the sensitivity analysis. However, the exclusion of Asians in the sensitivity analysis did result in significant enrichment of three EMT gene sets in obese CMS4 tumors that did not reach significance in the whole patient population. Taken together, these preliminary findings indicate CMS specific differences in obesity-regulated immune, metabolic, and stromal signaling gene set enrichment.

In addition to examining obesity-related transcriptomic differences, we examined transcriptomic differences between overweight vs. normal BMI patients for each CMS (Suppl. Fig 5A). Hallmark gene set enrichment with an FDR (p < 0.05) revealed immune-related gene set enrichment in CMS2 and CMS4. In contrast, gene set enrichment in CMS1 was strongly related to metabolic processes (heme metabolism, fatty acid metabolism, and bile acid metabolism). The only significant enrichment in CMS3 was a metabolic process gene set (oxidative phosphorylation). In a sensitivity analysis for the overweight to normal BMI comparison, we observed that all 21 enriched gene sets from the whole patient population were also significantly enriched when the population excluded Asians (Suppl. Fig. 5B). However, there were an additional 33 significantly enriched gene sets primary in Metabolism related gene sets (20 newly enriched) and Cell Cycle related gene set (9 newly enriched). In addition, 18 of the significantly enriched gene sets were observed in CMS3 tumors. These preliminary findings indicate that racial/ethnic differences may strongly influence transcriptomic differences in tumors from overweight patients. 

4) Comment 3 is critical to understand the impact of race/ethnicity on study results, due to substantial possible confounding. Please acknowledge this limitation also in the discussion.

We have added the following to the limitations section: A limitation of the current study is that it was only performed in the TCGA-COAD cohort which lacked racial diversity. The impact of race/ethnicity may have led to possible confounding. CMS categorization of CRC patients across the BMI categories led …

5) Comment 6 from reviewer 3 was either not understood or not done. Please add analyses removing stage IV, if adjustment is not possible.

We agree with the reviewer that weight loss is observed in a percentage of stage IV colon cancer patients prior to diagnosis. However, as reported by Walter et al (2016) and van Zutphen et al (2019), weight loss prior to diagnosis is also observed in stage I, II, and III CRC. According to Walter et al (2016), weight loss prior to diagnosis is more common in Stage II and III CRC than in stage IV. Therefore, removing stage IV colon cancer patients would not address the issue of weight loss prior to diagnosis. The lack of data on weight loss prior to diagnosis in our study is a limitation for all colon cancer stages. Therefore, we have revised the limitations in the Discussion section in the following manner to address this point: Additional limitations of the current study are that: 1) confounding factors that are risk factors for CRC such as smoking and alcohol consumption were not assessed; 2) weight loss prior to diagnosis which has been reported in all four colon cancer stages (93, 94), was not assessed; and 3) our findings in CMS1 tumors should be interpreted with caution because CMS1 is associated with familial disease and a younger patient population.

Walter V, Jansen L, Hoffmeister M, Ulrich A, Roth W, Bläker H, et al. Prognostic relevance of prediagnostic weight loss and overweight at diagnosis in patients with colorectal cancer. The American journal of clinical nutrition. 2016;104(4):1110-20.

van Zutphen M, Geelen A, Boshuizen HC, Winkels RM, Geijsen AJ, Wesselink E, et al. Pre-to-post diagnosis weight trajectories in colorectal cancer patients with non-metastatic disease. Supportive Care in Cancer. 2019;27(4):1541-9.

6) Reviewer 1 comment 3: The response starting with “The normal vs cancer analysis…” is meaningless and not relevant to the paper.

We apologize for the lack of relevance in the first sentence of the response. The first sentence in the response was not incorporated into the manuscript. So, there was nothing to edit in the current revision.

7) Please understand that CMS1 is most likely familial disease and patients are significantly younger. This needs to be added as a limitation, along with the racial composition.

We have revised the limitations to address the comment in following manner: Additional limitations of the current study are that confounding factors that are risk factors for CRC such as smoking and alcohol consumption were not assessed; weight loss prior to diagnosis which has been reported in all four stages (93, 94), was not assessed; and our findings in CMS1 tumors should be interpreted with caution because CMS1 is associated with familial disease and a younger patient population.

Despite these shortcomings of the study I remain enthusiastic of this publication. Please simply be mindful of the many limitations in the work.

---

## [Decision Letter · Decision Letter 2]

2 May 2022

Consensus molecular subtype differences linking colon adenocarcinoma and obesity revealed by a cohort transcriptomic analysis

PONE-D-21-24367R2

Dear Dr. Greene,

We’re pleased to inform you that your manuscript has been judged scientifically suitable for publication and will be formally accepted for publication once it meets all outstanding technical requirements.

Kind regards,

Katherine James, Ph.D.

Academic Editor

PLOS ONE

Additional Editor Comments (optional):

Reviewers' comments:

Reviewer's Responses to Questions

**Comments to the Author**

1. If the authors have adequately addressed your comments raised in a previous round of review and you feel that this manuscript is now acceptable for publication, you may indicate that here to bypass the “Comments to the Author” section, enter your conflict of interest statement in the “Confidential to Editor” section, and submit your "Accept" recommendation.

Reviewer #2: All comments have been addressed

2. Is the manuscript technically sound, and do the data support the conclusions?

Reviewer #2: Yes

3. Has the statistical analysis been performed appropriately and rigorously? 

Reviewer #2: Yes

4. Have the authors made all data underlying the findings in their manuscript fully available?

Reviewer #2: Yes

5. Is the manuscript presented in an intelligible fashion and written in standard English?

Reviewer #2: Yes

6. Review Comments to the Author

Reviewer #2: (No Response)

7. PLOS authors have the option to publish the peer review history of their article (what does this mean?). If published, this will include your full peer review and any attached files.

Reviewer #2: No

---

## [Editor Report · Acceptance letter]

6 May 2022

PONE-D-21-24367R2 

Consensus molecular subtype differences linking colon adenocarcinoma and obesity revealed by a cohort transcriptomic analysis 

Dear Dr. Greene:

I'm pleased to inform you that your manuscript has been deemed suitable for publication in PLOS ONE. Congratulations! Your manuscript is now with our production department. 

Kind regards, 

on behalf of

Dr. Katherine James 

Academic Editor

PLOS ONE